# COMPLEX LOGICAL INSTRUCTION GENERATION

## ABSTRACT

Instruction following has catalyzed the recent era of Large Language Models (LLMs) and is the foundational skill underpinning more advanced capabilities such as reasoning and agentic behaviors. As tasks grow more challenging, the logic structures embedded in natural language instructions becomes increasingly intricate. However, how well LLMs perform on such logic-rich instructions remains under-explored. We propose `LogicIFGen` and `LogicIFEval`. `LogicIFGen` is a **scalable**, **automated** framework for generating **verifiable** instructions from **code functions**, which can naturally express rich logic such as conditionals, nesting, recursion, and function calls. We further curate a collection of complex code functions and use `LogicIFGen` to construct `LogicIFEval`, a benchmark comprising 426 verifiable logic-rich instructions. Our experiments demonstrate that current state-of-the-art LLMs still struggle to correctly follow the instructions in `LogicIFEval`. Most LLMs can only follow fewer than 60% of the instructions, revealing significant deficiencies in their capacity to handle instructions that involve complex logical structures.

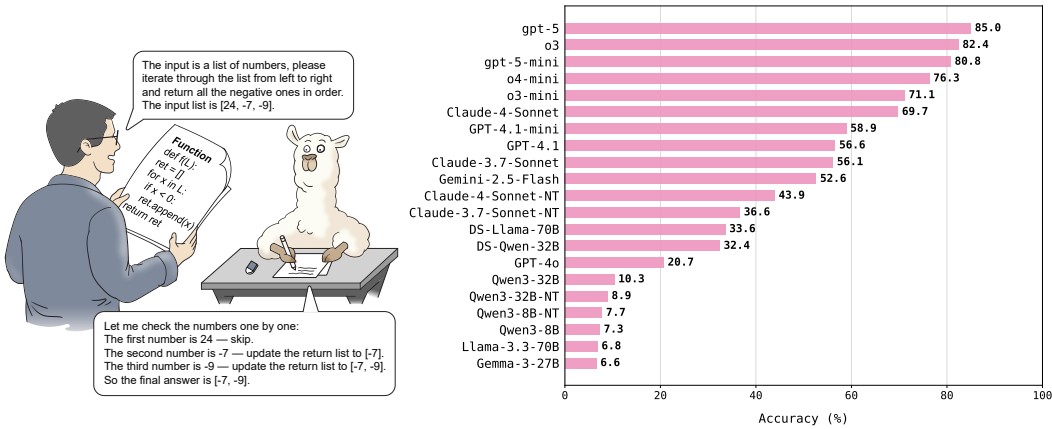

Figure 1: **(Left)** Instruction Following Test: LLMs are required to follow only the natural language instruction to simulate every logic of a code function via generating text. **(Right)** Overall instruction-following performance of evaluated models on `LogicIFEval`; NT denotes NoThinking. DS denotes DeepSeek.

## 1 INTRODUCTION

Before the advent of ChatGPT (Ouyang et al., 2022), chatbots built on earlier models such as GPT-1 (Radford & Narasimhan, 2018), GPT-2 (Radford et al., 2019), GPT-3 (Brown et al., 2020), and other architectures struggled to generate coherent and contextually appropriate utterances. At that time, it was difficult to imagine that such models could assist with tasks in daily life. With the emergence of instruction-following capabilities, large language models (LLMs) are now able to accurately understand basic human intentions and even leverage tools to perform a wide range

of tasks that enhance productivity, such as deep research[1], coding assistance[2], and scientific discovery (AI4Science & Quantum, 2023). The instructions for these tasks could contain rich logic structures such as sequencing, loops, nesting, recursion, and backtracking. Previous instruction-following evaluations typically focus on instructions with constraints on the response format (e.g., "fewer than 300 words") or content (e.g., "in Shakespeare's tone") (Zhou et al., 2023; Qin et al., 2024; Jiang et al., 2024) and seldom explore *how well LLMs perform on the instructions with rich logic structures*.

To address this, we first propose `LogicIFGen`, a **scalable**, **automated** framework for generating **verifiable** instructions from **code functions**, which can naturally contain rich logic structures. Given test input data, models are expected to rely **solely** on the natural language instruction to simulate every logic of the code function and produce the same results, analogous to being verbally guided by an examiner to process the input data step-by-step (see Figure 1 (Left)). The models are required to refrain from writing code or using external tools; instead, they must simulate and execute code logic only through text generation. This setting aligns more closely with instruction-following evaluation, as most tasks require the model to explicitly unfold the underlying logic. For example, when given an instruction such as "repeat asking the user for clarification until you fully understand the intent of the user," the model must perform each logical step through natural language alone. `LogicIFGen` obtains the reference labels by executing the code function on the same inputs. By comparing model outputs with these reference labels, we can easily verify whether a model follows the natural language instruction correctly. In addition, `LogicIFGen` incorporates state trackers to monitor intermediate logic flow, enabling us to double check whether models faithfully adhere to the instruction's internal logic rather than hallucinating the final results.

Second, we construct a benchmark called `LogicIFEval` using `LogicIFGen`, which contains 426 verifiable, logic-rich instructions paired with associated test cases. The functions used to generate `LogicIFEval` are the solutions of challenging simulation problems from competitive programming platforms, CodeForces[3] and POJ[4]. These simulation problem solutions are especially suitable for instruction-following evaluation because they require models to faithfully emulate complex, step-by-step processes and state transitions, often involving intricate control flow, edge case handling, and the coordination of multiple logic elements.

Experimental results show that most popular LLMs are only able to correctly follow fewer than 60% of the instructions in `LogicIFEval`, revealing a significant deficiency in their instruction-following ability (see Figure 1 (Right)). Open-source models continue to lag behind frontier models such as the OpenAI o-series and Anthropic Claude. As logical complexity increases, models find it increasingly difficult to accurately interpret and follow instructions. We also observe that incorporating explicit thinking before generating a response can potentially enhance instruction-following performance for large LLMs, but not for smaller LLMs. Further error analysis and case studies reveal key failure modes and highlight promising directions for advancing LLMs' ability to follow logic-rich instructions. We will release `LogicIFGen` and `LogicIFEval`, as well as a compute-friendly version of the benchmark, `LogicIFEval-mini`.

## 2 LOGICIFGEN: VERIFIABLE INSTRUCTION GENERATION FROM CODE

We introduce `LogicIFGen`, a framework that automatically generates verifiable, logic-rich instructions from **code functions**. Each instruction provides a comprehensive, step-by-step natural language description of the function's behavior, clearly specifying input/output formats and detailing all relevant control flows and data processing steps. Models are then expected to follow these instructions **without access to the source code** to process inputs and reproduce the function's outputs. Figure 2 illustrates our instruction generation framework with a running example. Additionally, `LogicIFGen` integrates *Multi-turn Difficulty Evolution* and *Multi-turn Verification and Refinement* modules, which dynamically adjust instruction complexity and verify correctness to ensure each instruction fully and accurately captures the function's logic. We provide a more detailed explanation of each module in the remainder of this section.

---

[1]https://openai.com/index/introducing-deep-research

[2]https://cursor.com

[3]https://codeforces.com

[4]http://poj.org

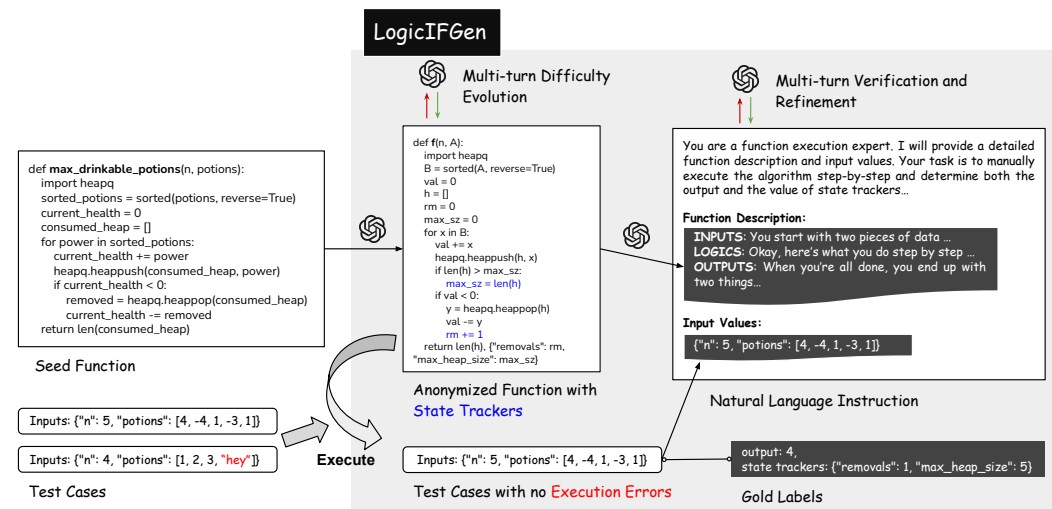

Figure 2: Pipeline of `LogicIFGen`. Given a seed function and its corresponding test cases, `LogicIFGen` generates natural language instructions along with gold labels, which include both the function outputs and the values of state trackers. 1) The input function is first anonymized and augmented with state trackers. 2) The anonymized function is then translated into a natural language description, producing a instruction that precisely describes its logic and expected behavior with test cases verified to have no execution errors. 3) Finally, the test cases are executed on the anonymized function to obtain the gold labels.

**Anonymized Function with State Trackers** The first step in `LogicIFGen` is to anonymize the original function so that **only its data operations and control logic remains**. `LogicIFGen` use an LLM to replace the function and variable names with simple, generic identifiers, single characters or common placeholders, thereby stripping away any semantic hints that could enable models to leverage domain-specific knowledge. Moreover, `LogicIFGen` asks an LLM to augment the function with **state trackers**, which are variables designed to log specific runtime states of the function, reflecting the function's internal logic and execution flow. These include, for example, the number of iterations of a `for` loop, the number of times an `if` block is executed, or the maximum length reached by a dynamically list. Models instructed via natural language are required to reproduce not only the function's outputs but also the values of state trackers. We consider the instruction to be correctly followed only when both the final output and the state tracker values match the ground truth. This ensures that models truly comprehend and execute the intended logic, rather than relying on shortcuts or hallucinating plausible outputs. The prompt used for this step is shown in Figure 7. Furthermore, to increase the logical complexity of the function, the *Multi-turn Difficulty Evolution* module in `LogicIFGen` prompts an LLM to evolve the code function by introducing more diverse compositions of logic units, such as loops, function calls, and recursion (see Appendix Figure 10 for the full prompt).

**Natural Instruction Generation** `LogicIFGen` uses an LLM (see Appendix Figure 8 for the prompt) to generate detailed natural language instructions based on the anonymized function with state trackers. The instructions adopt a conversational style (e.g., "Now you need to. . . ", "Next, go through. . . ") to guide step-by-step execution. Each instruction is crafted to be sufficiently precise so that an LLM can follow it to conduct all the data operations without access to the source code, producing identical outputs and state tracker values (as illustrated in Figure 1 (Left)). To ensure the instructions fully and correctly capture the sub-tasks and logic of the function, `LogicIFGen` incorporates the *Multi-turn Verification and Refinement* module. In this module, an LLM (see Appendix Figure 9 for the prompt) reviews the generated instructions, checking for comprehensive coverage of all code operations and flagging potential omissions, such as loop conditions, variable updates, or edge case handling. The instructions are iteratively refined based on this feedback until they accu-

rately reflect the complete operations and logic of the function. The full function description of of the running example in Figure 2 is as follows:

---

**Function Description**

**INPUTS:**
You start with two pieces of data. First is a single whole number called $n$, which tells you how many items you should expect. Second is a list called $A$, which contains exactly $n$ numeric values.

**LOGICS:**
Okay, here's what you do step by step. First, take your list $A$ and sort it from largest down to smallest to make a new list, which we'll call $B$. Now lay out four things on your workspace:

1. A running total called `val`, which you set to 0.

2. An empty collection called `h` – you can think of it as a bag where you'll always be able to pull out the smallest number.

3. A counter called `rm` for counting how many numbers you remove, which you set to 0.

4. A tracker called `max_sz` for the largest size `h` ever reaches, which you also set to 0.

Now you're going to process each number in $B$ in order, from the first (which is the largest) down to the last (the smallest). For each item, do the following:

- Let $x$ be the current item from $B$.

- Add $x$ to your running total `val` (so `val = val + x`).

- Drop $x$ into your bag `h`.

- Check how many items are now in `h`. If that count is bigger than `max_sz`, then write down the new count as the new `max_sz`.

- Next, look at your running total `val`. If `val` is now below zero (that means `val < 0`), you have to remove one item from `h`. To do that, find the smallest number currently sitting in `h`, call that $y$, and take it out of `h`. Then subtract $y$ from `val` (so `val = val - y`) and add 1 to your removal counter `rm`.

After you've done this for every single $x$ in $B$, you're finished with the loop. No more items left to process.

**OUTPUTS:**
When you're all done, you end up with two things. First, the number of items left in your bag `h` – that's the first result you'll report. Second, you have two statistics: the total `rm`, which tells you how many times you removed the smallest item, and `max_sz`, which tells you the largest number of items that ever piled up in `h` at once.

---

**Test Case Filtering and Gold Result Generation**   At this stage, we have two modalities representing the same operations and logic: the anonymized code function and the corresponding natural language instruction. Given the same input data, LLMs are expected to follow the instruction and produce results identical to those generated by the code function, including both the function outputs and the values of the state trackers. `LogicIFGen` generates these gold labels by executing the test cases on the anonymized function. Prior to this, it performs a preliminary execution step to filter out test cases that may trigger errors due to modifications introduced to function during the multi-turn difficulty evolution process. This ensures that only valid and executable test cases are retained for final gold label generation.

**Quantifiable Instruction Complexity**   One of the advantages of `LogicIFGen` is that the difficulty of the generated instructions is **quantifiable**: the anonymized function and natural language instruction describe the same operations and logic so we could use the code function as a proxy to analyze the difficulty of the instruction. We use Python's Abstract Syntax Tree (AST) package[5] to traverse the syntactic structure of each function and calculates several key measures. *Cyclomatic Complexity* ($C$) quantifies the total number of control flow decision points (such as if/elif statements, for/while loops, and try/except blocks), representing the number of linearly independent execution paths through the code. *Nesting depth* ($D$) tracks the maximum depth of nested control structures. *Function Call Count* ($F$) counts the number of function invocation, including both built-in and user-defined functions. *Function length* ($L$) is calculated as the number of lines spanned by the function definition. We adopt an intuitive weighting scheme for aggregating these AST-based measures based on their cognitive complexity, which has been proven effective in measure the logical complexity of codes (Muñoz Barón et al., 2020).

$$\text{Score} = D \times 3 + F \times 2 + C \times 1 + L \times 0.5. \tag{1}$$

Experiments in Section 4 also show the effectiveness of the intuitive weights. More advanced weighting methods like learning-based methods (Sepidband et al., 2025) could be explored in future.

---

[5]https://docs.python.org/3/library/ast.html

## 3  LOGICIFEVAL

In the previous section, we introduce `LogicIFGen`, which generates instructions and gold labels from code functions. However, we observed that the complexity of the seed function is crucial for producing challenging instructions. To this end, we curate a collection of simulation problem solutions and corresponding test cases from competitive programming platforms, including Codeforces and POJ. Specifically, for Codeforces, we select problems tagged with "implementation" and a difficulty score above 1700. For POJ, we include difficult simulation problems as identified by various online users. These functions require models to faithfully emulate complex, step-by-step processes and state transitions, involving intricate control flow, edge case handling, and the coordination of multiple logic elements, making them ideal sources for generating complex instructions used to evaluate the instruction-following ability. Then we construct `LogicIFEval` based on these functions as follows:

**Data Filtering**   We use a two-stage filtering process to make the seed functions more diverse and remove test cases not suitable for the instruction-following evaluation: *1) Seed Function Filtering*: We remove duplicate or highly similar functions to avoid redundancy. Specifically, we use OpenAI's text-embedding-3-small[6] model to compute embeddings of each function and calculate pairwise cosine similarities. If the similarity between two functions is greater than 0.7, we consider them near-duplicates. For each such pair, we keep only the longer function (measured by code length), since longer implementations often include richer logic and contribute to more challenging instruction-following tasks. *2) Test Case Filtering*: We remove test cases with unusual or problematic outputs to keep the dataset clean and manually executable. Specifically, we discard cases where: I) The values of state trackers are greater than or equal to 50 (to avoid overly large internal states). II) The output values have excessive precision (more than six decimal places). III) The state tracker dictionaries are malformed or incorrectly formatted. IV) The input values are too large (magnitude exceeding $10^7$). After this step, we remove functions with fewer than 3 test cases to ensure that each function has enough test coverage for reliable evaluation. These filtering steps are designed to prevent failures caused by large loops, deep recursion, or complex numerical computations that could overwhelm model capacity. This ensures that any model failures are attributable to weaknesses in instruction following, rather than the inherent difficulty of execution. The final dataset is easy to understand and execute step-by-step, staying consistent with our goal of generating clear and verifiable instructions. Originally, we collected 1,107 seed functions. After these filtering steps, 426 unique functions and 3,050 test cases remain.

**Data Generation and Human Verification**   We apply `LogicIFGen` using o4-mini as the generation LLM on the filtered seed functions and test cases, resulting in `LogicIFEval`, a benchmark comprising 426 complex instruction-following tasks. The numbers of turns used for Multi-turn Difficulty Evolution and Multi-turn Verification and Refinement are 1 and 3, respectively. Instructions that still fail verification after 3 turns are discarded. To evaluate the quality of the generated instructions, we hired five PhD-level experts in computer science (our co-authors) to conduct manual verification. The annotators were instructed to verify whether each line of the anonymized function is accurately and completely described in the corresponding natural language instruction. According to their assessment, 97% of the instructions fully and correctly capture the underlying function logic and their agreement is 97.79%, demonstrating the effectiveness of `LogicIFGen` in transforming code functions into natural language instructions. More details human evaluation process could be found in Appendix D. We further categorize the functions into difficulty levels using tercile-based thresholds derived from the complexity scores computed by Equation 1. The final benchmark contains 142 easy, 145 medium, and 139 hard instructions.

**Benchmark Statistics and Release**   The instructions in `LogicIFEval` have average of 3,428 characters and 662 words. The functions used to generate instructions have an average cyclomatic complexity of 11.10 and a maximum nesting depth of 3.16. In total, the benchmark includes 2,049 loops, 2,253 conditional statements, and 5,289 function calls, reflecting diverse control flow patterns. Each function is evaluated with an average of 7.2 test cases, resulting in 3,050 test cases overall. These statistics highlight the benchmark's scale and its focus on challenging instruction-following in LLMs. To support researchers with limited computational resources, we also release a representative

---

[6]https://platform.openai.com/docs/models/text-embedding-3-small

mini-benchmark, `LogicIFEval-mini`, which consists of 102 functions sampled in a stratified manner based on the complexity scores. Our experimental results in Section 4 show that constructing the min-benchmark based on the complexity scores is effective.

## 4  FRONTIER MODEL PERFORMANCE ON LOGICIFEVAL

| Model | Easy (142) | | | Medium (145) | | | Hard (139) | | | Average | | |
|---|---|---|---|---|---|---|---|---|---|---|---|---|
| | Output | State | **Both** | Output | State | **Both** | Output | State | **Both** | Output | State | **Both** |
| **Thinking Models** | | | | | | | | | | | | |
| gpt-5 | 94.37 | 95.07 | 90.85 | 97.24 | 91.03 | 89.66 | 88.49 | 82.01 | 74.10 | **93.43** | **89.44** | **84.98** |
| o3 | 94.37 | 90.14 | 89.44 | 93.10 | 87.59 | 84.83 | 84.89 | 79.86 | 72.66 | 90.85 | 85.92 | 82.39 |
| gpt-5-mini | 91.55 | 92.96 | 88.73 | 93.79 | 84.83 | 82.07 | 86.33 | 78.42 | 71.22 | 90.61 | 85.45 | 80.75 |
| o4-mini | 93.66 | 90.14 | 87.32 | 91.72 | 81.38 | 77.93 | 81.29 | 68.35 | 63.31 | 88.97 | 80.05 | 76.29 |
| o3-mini | 89.44 | 87.32 | 83.10 | 89.66 | 78.62 | 73.79 | 69.06 | 61.87 | 56.12 | 82.86 | 76.06 | 71.13 |
| Claude-4-Sonnet | 91.55 | 87.32 | 81.69 | 96.55 | 77.93 | 75.86 | 73.38 | 58.27 | 51.08 | 87.32 | 74.65 | 69.72 |
| Claude-3.7-Sonnet | 81.69 | 76.76 | 70.42 | 79.31 | 64.14 | 57.93 | 57.55 | 45.32 | 39.57 | 73.00 | 62.21 | 56.10 |
| Gemini-2.5-Flash | 79.58 | 75.35 | 72.54 | 64.14 | 55.17 | 53.10 | 41.01 | 34.53 | 31.65 | 61.74 | 55.16 | 52.58 |
| DS-Qwen-32B | 66.20 | 59.86 | 50.70 | 57.93 | 36.55 | 30.34 | 38.13 | 22.30 | 15.83 | 54.23 | 39.67 | 32.39 |
| DS-Llama-70B | 69.01 | 66.20 | 54.93 | 53.10 | 36.55 | 29.66 | 38.13 | 24.46 | 15.83 | 53.52 | 42.49 | 33.57 |
| Qwen3-32B | 38.73 | 31.69 | 23.94 | 17.93 | 5.52 | 3.45 | 13.67 | 4.32 | 3.60 | 23.47 | 13.85 | 10.33 |
| Qwen3-8B | 32.39 | 27.46 | 16.90 | 13.10 | 6.21 | 3.45 | 4.32 | 2.88 | 1.44 | 16.67 | 12.21 | 7.28 |
| **NoThinking Models** | | | | | | | | | | | | |
| GPT-4.1-mini | 85.92 | 81.69 | 74.65 | 82.07 | 64.83 | 60.00 | 71.22 | 49.64 | 41.73 | 79.81 | 65.49 | 58.92 |
| GPT-4.1 | 83.10 | 78.87 | 71.13 | 84.83 | 64.83 | 58.62 | 64.75 | 51.80 | 39.57 | 77.70 | 65.26 | 56.57 |
| Claude-4-Sonnet-NT | 80.28 | 76.06 | 69.01 | 63.45 | 46.21 | 37.93 | 44.60 | 30.22 | 24.46 | 62.91 | 50.94 | 43.90 |
| Claude-3.7-Sonnet-NT | 73.24 | 66.20 | 57.04 | 61.38 | 42.07 | 34.48 | 43.88 | 24.46 | 17.99 | 59.62 | 44.37 | 36.62 |
| GPT-4o | 59.86 | 49.30 | 38.73 | 32.41 | 24.14 | 17.93 | 12.95 | 10.07 | 5.04 | 35.21 | 27.93 | 20.66 |
| Qwen3-32B-NT | 33.10 | 29.58 | 19.72 | 18.62 | 9.66 | 4.14 | 9.35 | 4.32 | 2.88 | 20.42 | 14.55 | 8.92 |
| Llama-3.3-70B | 30.28 | 25.35 | 15.49 | 10.34 | 4.83 | 2.76 | 7.91 | 3.60 | 2.16 | 16.20 | 11.27 | 6.81 |
| Gemma-3-27B | 28.17 | 26.76 | 15.49 | 10.34 | 5.52 | 2.07 | 5.76 | 4.32 | 2.16 | 14.79 | 12.21 | 6.57 |
| Qwen3-8B-NT | 28.17 | 29.58 | 16.90 | 14.48 | 8.28 | 4.83 | 5.04 | 5.04 | 1.44 | 15.96 | 14.32 | 7.75 |

Table 1: Model performance (%) by complexity and average. "Output (State)" denotes a question is correct if the model produces the correct outputs (state trackers) for all associated test cases. "Both" denotes both the output and state trackers match. Overall performance are highlighted in pink. Performance of each difficulty level are in blue. The "Average" refers to micro-averaging, which is computed by summing the number of solved questions across the difficulty levels and dividing by the total number of questions.

We test 21 LLMs on `LogicIFEval`. **1) Frontier Thinking Models**: Models incorporating explicit thinking process before generating a response, including gpt-5, gpt-5-mini, o3, o4-mini, o3-mini, Claude-3.7-Sonnet and Claude-4-Sonnet, Gemini-2.5-Flash, Qwen3-32B and Qwen3-8B, DeepSeek-R1-Distill(DS)-Llama-70B and DS-Qwen-32B. **2) Frontier NoThinking (NT) Models**: Models directly give a response without explicit thinking: GPT-4.1, GPT-4.1-mini, GPT-4o, Claude-3.7-Sonnet-NT and Claude-4-Sonnet-NT, Gemma-3-27B, Qwen3-32B-NT and Qwen3-8B-NT, and Llama-3.3-70B(-Instruct).

**Inference Setting**    For closed-source models, we use the default temperature setting provided by the respective API. For open-source models, we adopt the official recommended inference settings; for example, for the Qwen3 series reasoning models, the temperature is set to 0.6. If no specific recommendation is available, we use a temperature of 1.0 by default. For all models, we set the maximum number of generated tokens to 16k to ensure that the models have sufficient capacity to follow instructions and process the inputs.

**Main Results**   A model is considered to successfully follow a natural language instruction if it passes all associated test cases by producing both the correct outputs and accurate state tracker values. Figure 1 and the last column of Table 1 report the overall accuracies across evaluated models. **The top-performing models are the OpenAI gpt-5, o-series and Claude-4-Sonnet**, with the best-performing model, gpt-5, achieving an accuracy of 84.98%. These results highlight the strong instruction-following and logic execution capabilities of advanced proprietary LLMs. In contrast, GPT-4o achieves only 20.66%, underperforming relative to other OpenAI models. **Besides, widely used open-source models still lag significantly behind.** For example, Qwen3-32B, Gemma-3-27B, and Llama-3.3-70B all score below 11%, failing to correctly execute the multi-step logic and state tracking required by `LogicIFEval`, which highlights a clear performance gap between commercial and open-source LLMs on this benchmark. We also notice that **explicit thinking before response can potentially improve instruction following for large LLMs.** For instance, Claude-4-Sonnet achieves 69.72% accuracy, notably outperforming Claude-4-Sonnet-NT (43.9%) and Claude-3.7-Sonnet (56.1%) outperforms Claude-3.7-Sonnet-NT (36.62%). Similarly, OpenAI's thinking models, gpt5 and o-series, perform substantially better than other non-thinking models from OpenAI. However, the Qwen3 variants show little difference between thinking and non-thinking modes, suggesting that explicit thinking helps only when the underlying model has sufficiently strong capabilities. We also report model performance on `LogicIFEval-mini` in Figure 6. The results show that models exhibit nearly identical rankings compared to the full benchmark, indicating that `LogicIFEval-mini` effectively represents the whole benchmark and preserves a similar distribution.

**Results Across Difficulty Levels**   Table 1 presents model performance across three difficulty levels in `LogicIFEval`, with separate evaluations for output correctness, state tracker correctness, and their intersection ("Both"). We could see that **all models show a clear degradation in performance as difficulty increases** (see blue columns ), validating the effectiveness of our AST-based complexity scoring strategy. For instance, gpt-5 drops from 90.85% on Easy tasks to 74.10% on Hard ones, and GPT-4.1-mini from 82.07% to 41.73%. This trend confirms that our benchmark's stratification meaningfully reflects logical complexity of instructions. **Second, output accuracy consistently exceeds state tracker accuracy across nearly all models**, especially as complexity increases. For example, GPT-4.1-mini achieves 71.22% output accuracy on Hard instructions, but only 49.64% in state tracking. This implies that models may generate correct answers without strictly adhering to the intended logic steps or may follow alternative logic paths. These observations underscore the importance of adding state trackers to supervise the logic flow in complex instruction-following tasks.

## 5   ANALYSIS

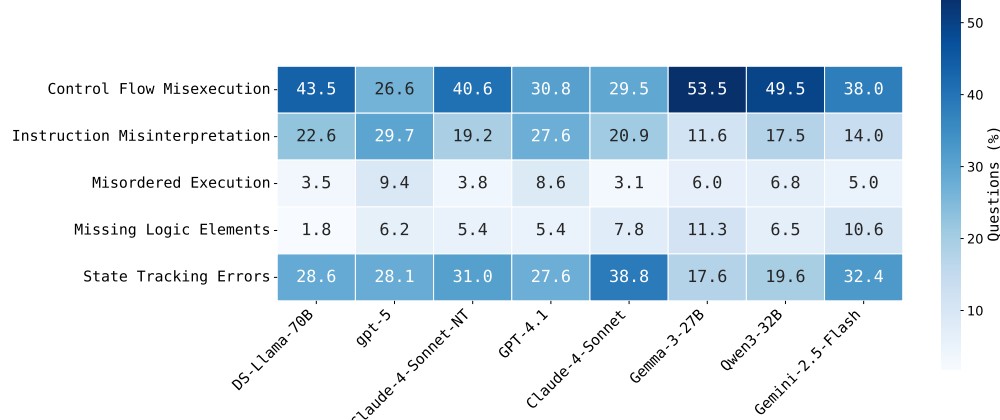

Figure 3: Error Type Distribution in Test Cases

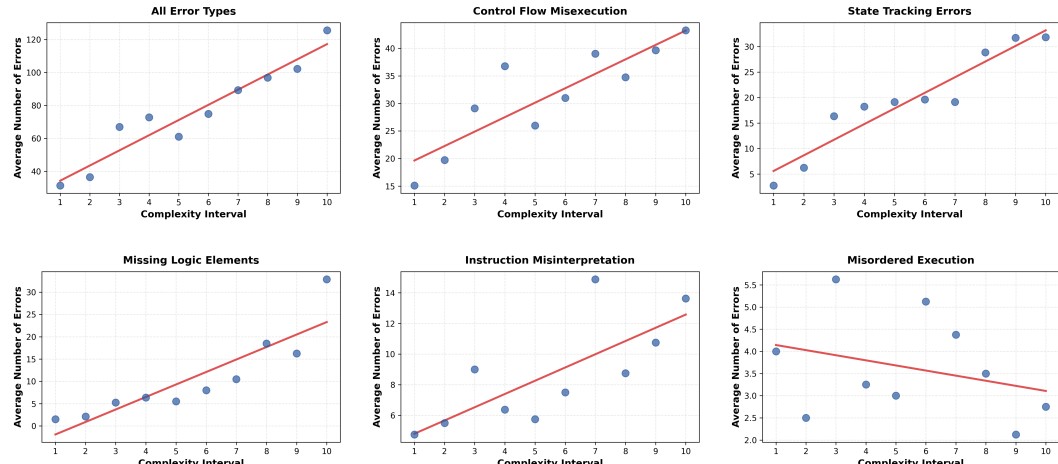

Figure 4: Error distribution across complexity intervals. Blue points represent average error counts across 8 models at each complexity interval, with red lines indicating linear trends. The top-left panel shows all error types combined, while the other panels show individual failure modes.

**Failure Modes** To understand the reasons of why models can not pass the test cases, we first go through some errors and summarize them as following five types: **1) Control Flow Misexecution**: incorrect, incomplete, or inconsistent execution of core control structures (e.g., loops, branches, function calls), including wrong iteration counts, improper branching, or mishandled recursion/returns. **2) State Tracking Errors**: failure to correctly maintain or update internal variables or data structures, such as counters, flags, arrays, stacks, or accumulated values. **3) Missing Logic Elements**: omission of required components (e.g., loops, branches, edge case handling or initialization). **4) Misordered Execution**: performing steps in the wrong sequence, such as using uninitialized variables, premature function calls, or out-of-order updates. **5) Instruction Misinterpretation**: misunderstanding the instruction's intent, leading to hallucinated steps, misapplied patterns, or ignored constraints. We map the error cases to these categorizes with the help of GPT-4.1.

Figure 3 summarizes the error types made by some popular models. We could see that the most frequent error categories across models are Control Flow Misexecution, Instruction Misinterpretation, and State Tracking Errors. In contrast, Missing Logic Elements and Misordered Execution are consistently low (mostly under 10%), suggesting that **most models can identify the required logic components ("what to do") and their approximate ordering ("when to do it"). However, they often struggle with actually executing the logic elements correctly**, either by mismanaging control structures (e.g., loop iterations, function calls), hallucinating or misinterpreting instruction details, or failing to track internal state variables accurately over time. In addition, **open-source models like Qwen3-32B, DS-Llama-70B, and Gemma-3-27B exhibit especially high rates of Control Flow Misexecution**, up to 53.5% in the case of Gemma-3-27B. This highlights their difficulty in faithfully reproducing the logic-heavy instruction steps, which require consistent handling of nesting, conditionals, and function boundaries. Besides, **State Tracking Errors is also a major issue across nearly all models**. For instance, Claude-4-Sonnet and Gemini-2.5-Flash show 38.8% and 32.4% error rates respectively, indicating frequent failures in maintaining correct variable states when following the instructions. These include losing track of counters, failing to propagate updates through data structures, or resetting intermediate results incorrectly. This reinforces the importance of evaluating beyond output correctness. We show representative cases for each error type in Figure 5 and the full examples in Appendix B. It should be noted that the error types are not strictly mutually exclusive. For example, the missing logic elements error observed in the fourth example of Figure 5 can also be attributed to the incorrect execution of the while condition.

To analyze the relationship between error rate and logical complexity, we divide the instructions in `LogicIFEval` into ten intervals based on their complexity scores computed using Eq. 1. Interval 1 corresponds to the easiest instructions, while Interval 10 contains the most difficult ones. For each interval, we compute the average number of error cases across all eight representative models and visualize the trend in Figure 4. **The results show a clear positive correlation between log-**

**ical complexity and error frequency: as complexity increases, models produce substantially more errors**. In particular, the proportions of Control Flow Misexecution, Instruction Misinterpretation, Missing Logic Elements, and State Tracking Errors rise sharply, suggesting that current LLMs mainly struggle with understanding and executing complex logical structures. In contrast, Misordered Execution remains consistently low across all complexity levels, indicating that models generally follow the execution order specified in the instructions.

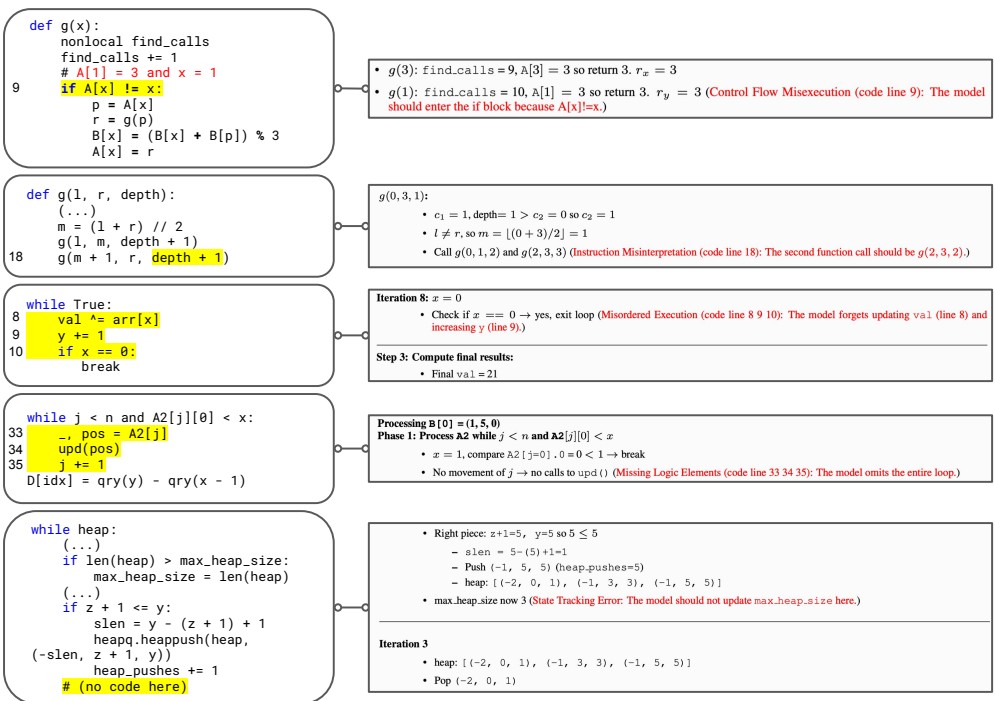

Figure 5: Error Cases: On the left are the **excerpts** from function codes where the model makes errors. On the right are **excerpts** from the LLMs' responses, highlighting their failures across different modes. The explanations for model failures are indicated in red, and the corresponding code lines are highlighted. Please note that the model only has access to the natural language instruction, which could correctly describe the logic, when solving the tasks; the code is provided here solely to facilitate understanding of the errors.

**Why Thinking Helps Large LLMs** As indicated in Section 4, incorporating explicit thinking before generating a response can potentially enhance instruction-following performance for large LLMs. To gain deeper insights into this effect, we conduct a case analysis focusing on instances where Claude-4-Sonnet produces correct results, whereas Claude-4-Sonnet-NT fails. These cases can be categorized into two types: 1) the model arrives at the correct answer during the thinking process itself, and 2) the model articulates a detailed, step-by-step plan to solve the task. Both scenarios suggest that explicit thinking encourages the model to slow down and solve the problem more deliberately, rather than relying on pattern matching or intuitive leaps. We provide two representative examples in the Appendix E due to space limit.

## 6 RELATED WORK

**Instruction Following** General instruction-following evaluation work typically focuses on instructions that impose constraints on the response format or content (Zhou et al., 2023; Qin et al., 2024; Pyatkin et al., 2025). These research centers on enabling models to follow instructions with multiple

constraints (Jaroslawicz et al., 2025; Pyatkin et al., 2025). Wen et al. (2024) investigates compositional constraint following, introducing logic structures such as sequential logic and branching logic. In comparison, the logic in our benchmark are significantly more complex and diverse. Verification methods generally fall into two categories: heuristic functions or LLM-as-judge. While LLM-as-judge has been shown to correlate highly with human judgments (Qin et al., 2024), there is still room for improvement (Zeng et al., 2023). In addition to general-purpose instruction-following benchmarks, other datasets target specific scenarios or constraints, such as length control (Zhang et al., 2025), long-context settings (Wu et al., 2024), or agentic scenarios (Qi et al., 2025; Li et al., 2025b). Yang et al. (2025) explore LLMs' ability to adhere to user intent while producing functionally accurate code. In contrast, our work uses code as the source to generate instructions, requiring LLMs to generate text, rather than code, in response. Some studies have suggested that reasoning may decrease instruction-following performance (Li et al., 2025a; Fu et al., 2025). However, our findings in Section 4 indicate that reasoning can actually enhance instruction-following for logic-rich instructions. Further research like Tam et al. (2024); Qin et al. (2025) is needed to clarify the relationship between reasoning and instruction following. To the best of our knowledge, we are the first to investigate whether LLMs can precisely follow logic-rich instructions and how to generate such instructions in scale.

**Code Execution** The work on code execution evaluates LLMs by giving them code snippets and asking them to act as interpreters that predict execution outcomes La Malfa et al. (2024); Gu et al. (2024); Jain et al. (2024); La Malfa et al. (2025); Sun et al. (2025). In contrast, our goal is to assess whether LLMs can follow complex logical instructions expressed in natural language. Code functions in our framework are not the input modality; they serve only as a scaffolding to synthesize natural-language instructions with rich logical structures. Models are then evaluated on whether they can faithfully execute these instructions step-by-step in natural language. This capability is of broad practical relevance, as modern agentic workflows and system constraints are primarily communicated in natural language and often encode intricate conditional reasoning, error handling, or function-calling behaviors. Our benchmark directly targets this ability.

## 7 CONCLUSION AND FUTURE WORK

In this paper, we introduce `LogicIFGen`, a framework that automatically generates verifiable, task-intensive, logic-rich instructions from code functions. We also present `LogicIFEval`, a challenging instruction-following evaluation benchmark constructed using `LogicIFGen`, which consists of 426 tasks featuring complex logic. Experiments show that most proprietary and open-source models struggle to solve these tasks, revealing a significant deficiency in instruction-following capabilities. Future work includes exploring the use of `LogicIFGen` as a verifiable instruction generator for model training and `LogicIFEval` for evaluation to develop models with generalized and robust instruction-following capabilities. We believe that as LLMs are deployed in diverse agentic tasks, instruction following is the most crucial, yet often overlooked, capability to handle the growing complexity of memory and context managing (Gutiérrez et al., 2024; Khattab et al., 2023), tool calling (Wu et al., 2025; Yin et al., 2025), reasoning, planning, and acting (Liu et al., 2025).

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

# A LOGICIFEVAL-MINI PERFORMANCE

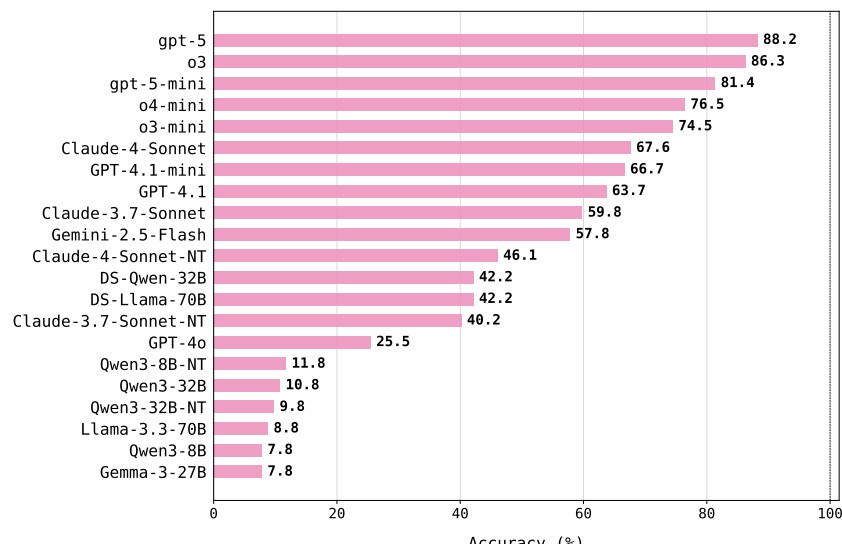

Figure 6: Overall instruction-following performance (%) of all evaluated models on LogicIFEval-mini, measured by the percentage of test cases where both the output and state trackers match the ground truth. Models are sorted by overall accuracy.

# B    ERROR CASES

## B.1    ERROR CASE: POJ-1852

**Error Type:** Control Flow Misexecution

**Test Input:**

```
1  4, [[2, 1, 2], [2, 2, 3], [1, 1, 3], [2, 3, 1], [1, 4, 4]]
```

**LLM Results:**

```
1  output: 1
2  stats: {'find_calls': 12, 'unions': 2}
```

**Code Results:**

```
1  output: 1
2  stats: {'find_calls': 13, 'unions': 2}
```

**Function:**

```python
1  def f(n, L):
2      A = list(range(n+1))
3      B = [0]*(n+1)
4      find_calls = 0
5      unions = 0
6      def g(x):
7          nonlocal find_calls
8          find_calls += 1
9          if A[x] != x:
10             p = A[x]
11             r = g(p)
12             B[x] = (B[x] + B[p]) % 3
13             A[x] = r
14         return A[x]
15     m = 0
16     for d, x, y in L:
17         if x < 1 or x > n or y < 1 or y > n:
18             m += 1
19             continue
20         r = 0 if d == 1 else 1
21         rx = g(x)
22         ry = g(y)
23         if rx == ry:
24             if (B[x] - B[y] - r) % 3 != 0:
25                 m += 1
26         else:
27             unions += 1
28             delta = (B[y] - B[x] + r) % 3
29             A[rx] = ry
30             B[rx] = delta
31     return m, {'find_calls': find_calls, 'unions': unions}
```

---

**Function Description**

**INPUTS:** You start with a single integer n and a list L. The list L contains multiple items, and each item is a triple of integers written as $(d, x, y)$. Those are all the pieces you'll need to begin.

**LOGICS:** First, what you're going to do is build two lists of length $n + 1$. Call the first one A and fill it so that $A[0] = 0$, $A[1] = 1$, $A[2] = 2$, all the way up to $A[n] = n$. Call the second one B and fill every slot from $B[0]$ through $B[n]$ with zero. Then make two counters and set both to zero: one called find_calls and the other called unions.

Next you define a little helper routine, let's say it's called g(x). Here's exactly how you run g on some index $x$:

1.  Right when you enter g, add 1 to find_calls.

2.  Check if $A[x]$ is the same number as $x$. If it is, you've found a root and you just return $x$ immediately.

3.  If $A[x]$ is not $x$, that means $x$ points up to a parent. So write down $p = A[x]$. Then call g(p) to chase up the chain—which itself bumps find_calls again and eventually returns a root $r$.

4.  After g(p) returns $r$, you adjust $B[x]$: take the old $B[x]$, add $B[p]$, then reduce that sum modulo 3 (so you do $(B[x] + B[p]) \bmod 3$) and store it back into $B[x]$.

5.  Also update $A[x]$ so that it now directly equals $r$.

6.  Finally return $r$.

All set up now? Good. Next you create a counter m and set it to zero. This will count how many triples get tossed or prove contradictory. Then you go through each triple in your list L in order. For each triple do this:

- First look at $x$ and $y$ and see if either one is below 1 or above $n$. If $x < 1$ or $x > n$ or $y < 1$ or $y > n$, add 1 to m and skip the rest of these steps for that triple. (That's just a bad-index check.)

- Otherwise, you figure out a small value $r$: if $d$ equals 1 then $r = 0$, else $r = 1$.

- Then call g(x) and call g(y). Store the results as rx and ry. Those are the roots of $x$ and $y$, and by now find_calls has grown by two.

- Now check: if rx and ry are the same number, that means $x$ and $y$ are already in the same group. In that case you compute $(B[x] - B[y] - r) \bmod 3$; if that result is not zero, it means this triple contradicts existing info, so add 1 to m. If it's zero, do nothing extra—just move on to the next triple.

- If rx and ry are different, you need to merge the two groups. So do unions $+ = 1$. Then compute $\delta = (B[y] - B[x] + r) \bmod 3$. After that, link the root rx under ry by setting A[rx] = ry, and store B[rx] = $\delta$. That finishes merging for this triple.

- Then move on to the next triple until you exhaust the list.

When you've processed every triple in L, you stop. The number m reflects how many triples were either invalid or contradictory, find_calls tells you how many times you ran the helper g, and unions tells you how many merges you performed.

**OUTPUTS:** At the end you get two things. First you get the integer m, which is the count of all the bad or contradictory triples you hit. Second you get a small record or dictionary with two entries: 'find_calls' is the total number of times you entered the helper g, and 'unions' is how many times you merged two distinct roots.

## LLM Reasoning

**Reasoning:**
Let me work through this algorithm step by step.

**Initialization:**

- n = 4, L = [[2, 1, 2], [2, 2, 3], [1, 1, 3], [2, 3, 1], [1, 4, 4]]
- A = [0, 1, 2, 3, 4] (indices 0 through 4)
- B = [0, 0, 0, 0, 0] (all zeros)
- find_calls = 0, unions = 0, m = 0

**Processing Triple 1: [2, 1, 2]**

- $d = 2$, $x = 1$, $y = 2$. Both in bounds $[1, 4]$.
- $r = 1$ (since $d \neq 1$)
- $g(1)$: find_calls = 1, A[1] = 1 so return 1. $r_x = 1$
- $g(2)$: find_calls = 2, A[2] = 2 so return 2. $r_y = 2$
- $r_x \neq r_y$, so merge: unions = 1
- delta = $(B[2] - B[1] + 1) \bmod 3 = (0 - 0 + 1) \bmod 3 = 1$
- A[1] = 2, B[1] = 1

**Processing Triple 2: [2, 2, 3]**

- $d = 2$, $x = 2$, $y = 3$. Both in bounds.
- $r = 1$
- $g(2)$: find_calls = 3, A[2] = 2 so return 2. $r_x = 2$
- $g(3)$: find_calls = 4, A[3] = 3 so return 3. $r_y = 3$
- $r_x \neq r_y$, so merge: unions = 2
- delta = $(B[3] - B[2] + 1) \bmod 3 = (0 - 0 + 1) \bmod 3 = 1$
- A[2] = 3, B[2] = 1

**Processing Triple 3: [1, 1, 3]**

- $d = 1$, $x = 1$, $y = 3$. Both in bounds.
- $r = 0$
- $g(1)$: find_calls = 5, A[1] = 2 $\neq$ 1, so p = 2
  - $g(2)$: find_calls = 6, A[2] = 3 $\neq$ 2, so p = 3
    - $g(3)$: find_calls = 7, A[3] = 3 so return 3
  - B[2] = $(1 + 0) \bmod 3 = 1$, A[2] = 3, return 3
  - B[1] = $(1 + 1) \bmod 3 = 2$, A[1] = 3, return 3
- $r_x = 3$
- $g(3)$: find_calls = 8, A[3] = 3 so return 3. $r_y = 3$
- $r_x = r_y = 3$, check consistency: $(B[1] - B[3] - 0) \bmod 3 = (2 - 0 - 0) \bmod 3 = 2 \neq 0$
- Contradiction! m = 1

**Processing Triple 4: [2, 3, 1]**

- $d = 2$, $x = 3$, $y = 1$. Both in bounds.
- $r = 1$
- $g(3)$: find_calls = 9, A[3] = 3 so return 3. $r_x = 3$
- $g(1)$: find_calls = 10, A[1] = 3 so return 3. $r_y = 3$ (Control Flow Misexecution (code line 9): The model should enter the if block because A[x]!=x.)
- $r_x = r_y = 3$, check consistency: $(\text{B}[3] - \text{B}[1] - 1) \bmod 3 = (0 - 2 - 1) \bmod 3 = (-3) \bmod 3 = 0$
- Consistent, no increment to m

**Processing Triple 5: [1, 4, 4]**

- $d = 1$, $x = 4$, $y = 4$. Both in bounds.
- $r = 0$
- $g(4)$: find_calls = 11, A[4] = 4 so return 4. $r_x = 4$
- $g(4)$: find_calls = 12, A[4] = 4 so return 4. $r_y = 4$
- $r_x = r_y = 4$, check consistency: $(\text{B}[4] - \text{B}[4] - 0) \bmod 3 = (0 - 0 - 0) \bmod 3 = 0$
- Consistent, no increment to m

**Output:** 1

**Statistics:** $\{$'find_calls': 12, 'unions': 2$\}$

## B.2 ERROR CASE: CODEFORCES-328A

**Error Type:** Instruction Misinterpretation

**Test Input:**

```
1  4, [1, 0, 1, 0]
```

**LLM Results:**

```
1  output: 3
2  stats: {'rec_calls': 7, 'max_depth': 5}
```

**Code Results:**

```
1  output: 3
2  stats: {'rec_calls': 7, 'max_depth': 3}
```

**Function:**

```
1  def f(n, arr):
2      c1 = 0
3      c2 = 0
4      x = 0
5      for val in arr:
6          x += val
7      A = [1 if val == 0 else -1 for val in arr]
8      def g(l, r, depth):
9          nonlocal c1, c2
10         c1 += 1
11         if depth > c2:
12             c2 = depth
13         if l == r:
14             v = A[l]
15             return v, v, v, v
16         m = (l + r) // 2
17         lt, lp, ls, lb = g(l, m, depth + 1)
18         rt, rp, rs, rb = g(m + 1, r, depth + 1)
19         total = lt + rt
20         pref = max(lp, lt + rp)
21         suf = max(rs, rt + ls)
22         best = max(lb, rb, ls + rp)
23         return total, pref, suf, best
24     _, _, _, b = g(0, n - 1, 1)
25     return x + b, {'rec_calls': c1, 'max_depth': c2}
```

---

**Function Description**

**INPUTS:** The inputs are a number called n and a list called arr containing n numbers.

**LOGICS:** Start by writing down three counters named $c_1$, $c_2$, and x and set them all to zero. Now go through each value in arr one at a time and add that value to x. When you finish, x holds the sum of arr.

Next, build a new list called A: begin with it empty, then for every element in arr in order, if the element is zero append 1 into A, otherwise append −1.

Now you are going to define a process named g that takes three numbers l, r, and depth. Every time you start g, do the following: first add 1 to $c_1$, then compare depth with $c_2$ and if depth is larger replace $c_2$ with depth. Then check if l equals r. If they are equal, look up A at index l, call that value v, and return four copies of v.

If l is less than r, compute m as the floor of $(l + r)$ divided by 2. Now call g on the left half by using l, m, and depth plus 1, and write down its four outputs as lt, lp, ls, and lb in that order. Then call g on the right half by using m + 1, r, and depth plus 1, and write down its four outputs as rt, rp, rs, and rb.

After you have those eight numbers, calculate total as lt plus rt, then pref as whichever is larger between lp and lt plus rp, then suf as whichever is larger between rs and rt plus ls, and finally best as the largest of lb, rb, and ls plus rp. Then return total, pref, suf, and best from this call.

Once you have defined that, kick off the process by calling g with l = 0, r = n minus 1, and depth = 1. When it finishes, ignore the first three values it gives you and call the fourth value b. At this point $c_1$ is the total count of g calls, and $c_2$ is the deepest level you reached. Finally, add b to x and that gives you the main result.

**OUTPUTS:** You will end up with two things: a number equal to x plus b, and a small object with two fields, rec_calls set to $c_1$ (the total number of g calls) and max_depth set to $c_2$ (the deepest recursion level you reached).

### LLM Reasoning

**Reasoning:**
Let me work through this algorithm step by step.

**Step 1: Initialize variables**

- $n = 4$, arr $= [1, 0, 1, 0]$
- $c_1 = 0, c_2 = 0, x = 0$

**Step 2: Calculate sum of arr**
Going through each value in arr and adding to $x$:

- $x = 0 + 1 = 1$
- $x = 1 + 0 = 1$
- $x = 1 + 1 = 2$

- $x = 2 + 0 = 2$

So $x = 2$

**Step 3: Build array $A$**
For each element in arr: if 0 append 1, otherwise append $-1$:

- arr$[0] = 1 \rightarrow$ append $-1$
- arr$[1] = 0 \rightarrow$ append $1$
- arr$[2] = 1 \rightarrow$ append $-1$

- arr$[3] = 0 \rightarrow$ append $1$

So $A = [-1, 1, -1, 1]$

**Step 4: Execute $g(0, 3, 1)$**

$g(0, 3, 1)$:

- $c_1 = 1$, depth$= 1 > c_2 = 0$ so $c_2 = 1$
- $l \neq r$, so $m = \lfloor (0 + 3)/2 \rfloor = 1$
- Call $g(0, 1, 2)$ and $g(2, 3, 3)$ (Instruction Misinterpretation (code line 18): The second function call should be $g(2, 3, 2)$.)

$g(0, 1, 2)$:

- $c_1 = 2$, depth$= 2 > c_2 = 1$ so $c_2 = 2$
- $l \neq r$, so $m = \lfloor (0 + 1)/2 \rfloor = 0$
- Call $g(0, 0, 3)$ and $g(1, 1, 4)$

$g(0, 0, 3)$:

- $c_1 = 3$, depth$= 3 > c_2 = 2$ so $c_2 = 3$
- $l = r = 0$, so $v = A[0] = -1$
- Return $(-1, -1, -1, -1)$

$g(1, 1, 4)$:

- $c_1 = 4$, depth$= 4 > c_2 = 3$ so $c_2 = 4$
- $l = r = 1$, so $v = A[1] = 1$
- Return $(1, 1, 1, 1)$

Back to $g(0, 1, 2)$:
$$\begin{cases} lt = -1, & lp = -1, & ls = -1, & lb = -1; \\ rt = 1, & rp = 1, & rs = 1, & rb = 1 \end{cases}$$
$$\text{total} = -1 + 1 = 0$$
$$\text{pref} = \max(-1, -1 + 1) = 0$$
$$\text{suf} = \max(1, 1 + (-1)) = 1$$
$$\text{best} = \max(-1, 1, -1 + 1) = 1$$

Return $(0, 0, 1, 1)$

$g(2, 3, 3)$:

- $c_1 = 5$, depth$= 3 \leq c_2 = 4$ so $c_2$ stays 4
- $l \neq r$, so $m = \lfloor (2 + 3)/2 \rfloor = 2$
- Call $g(2, 2, 4)$ and $g(3, 3, 5)$

$g(2, 2, 4)$:

- $c_1 = 6$, depth$= 4 = c_2 = 4$ so $c_2$ stays 4
- $l = r = 2$, so $v = A[2] = -1$
- Return $(-1, -1, -1, -1)$

$g(3, 3, 5)$:

- $c_1 = 7$, depth$= 5 > c_2 = 4$ so $c_2 = 5$
- $l = r = 3$, so $v = A[3] = 1$
- Return $(1, 1, 1, 1)$

Back to $g(2, 3, 3)$:

$$\begin{cases} lt = -1, & lp = -1, & ls = -1, & lb = -1; \\ rt = 1, & rp = 1, & rs = 1, & rb = 1 \end{cases}$$

$$\text{total} = -1 + 1 = 0$$
$$\text{pref} = \max(-1, -1 + 1) = 0$$
$$\text{suf} = \max(1, 1 + (-1)) = 1$$
$$\text{best} = \max(-1, 1, -1 + 1) = 1$$

Return $(0, 0, 1, 1)$

Back to $g(0, 3, 1)$:

$$\begin{cases} lt = 0, & lp = 0, & ls = 1, & lb = 1; \\ rt = 0, & rp = 0, & rs = 1, & rb = 1 \end{cases}$$

$$\text{total} = 0 + 0 = 0$$
$$\text{pref} = \max(0, 0 + 0) = 0$$
$$\text{suf} = \max(1, 0 + 1) = 1$$
$$\text{best} = \max(1, 1, 1 + 0) = 1$$

Return $(0, 0, 1, 1)$

**Step 5: Final calculations**

- $b = 1$ (fourth value from $g$)
- $c_1 = 7$ (total $g$ calls)
- $c_2 = 5$ (max depth)
- Final result $= x + b = 2 + 1 = 3$

**Output:** 3

**Statistics:** $\{'\texttt{rec\_calls}': 7, '\texttt{max\_depth}': 5\}$

### B.3 ERROR CASE: CODEFORCES-2018B

**Error Type:** Misordered Execution

**Test Input:**

```
1  8, [13, 27, 35, ...] (len=8)
```

**LLM Results:**

```
1  output: 21
2  stats: {'xor_operations': 7, 'and_operations': 7}
```

**Code Results:**

```
1  output: 24
2  stats: {'xor_operations': 8, 'and_operations': 7}
```

**Function:**

```
1  def f(n, arr):
2      m = n - 1
3      val = 0
4      x = m
5      y = 0
6      z = 0
7      while True:
8          val ^= arr[x]
9          y += 1
10         if x == 0:
11             break
12         x = (x - 1) & m
13         z += 1
14     return val, {'xor_operations': y, 'and_operations': z}
```

**Function Description**

**INPUTS:** You're working with two pieces of data: a number called n, and a list called `arr` that contains exactly n numeric entries, indexed from 0 up to n minus one.

**LOGICS:** Start by creating a variable called m by taking n and subtracting 1. Next, create a running total named `val` and set it to zero. Now set up another variable x and give it the same value as m. Also prepare two counters, y and z, and initialize both of them at zero. What you're going to do now is enter a loop that keeps going until x becomes zero. Every time you go through this loop, follow these exact steps:

1. First, look up the item at index x in your `arr` list and apply a bitwise exclusive OR between that item and your current `val`. To do that by hand, write both numbers in binary, compare each bit pair, write down a 1 if the bits differ or a 0 if they match, then turn that resulting bit pattern back into a number and call it your new `val`.

2. Once you've done that, add 1 to your counter y to record that XOR operation.

3. Next, check whether x is equal to zero. If x is zero, break out of the loop immediately—you're done.

4. If x is not zero, you need to update x for the next round: subtract 1 from your current x, then take that result and apply a bitwise AND with m (again, line up the binary bits of (x-1) and m, write down a 1 only where both bits are 1, then convert that back into a number). Store that outcome back into x, and because you just performed an AND, add 1 to your counter z.

5. Then go back to the top of the loop and repeat exactly those steps—XOR and increment y, check for x == 0 and maybe break, AND and increment z—until x finally reaches zero and you exit.

**OUTPUTS:** When you're finished, you'll have two things: the first is the final number stored in `val` after all the XOR steps, and the second is a small dictionary showing {'xor_operations': y, 'and_operations': z} which equals your y counter (the total number of XORs you performed), and z counter (the total number of ANDs you performed).

**LLM Reasoning**

**Reasoning:**
Given the input value: $n = 8$, and `arr` = [13, 27, 35, 49, 59, 61, 73, 83].

**Step 1: Initialization**

- Compute $m = n - 1 = 8 - 1 = 7$ (binary: 0111)
- Initialize `val` $= 0$
- $x = m = 7$
- $y$ (XOR counter) $= 0$
- $z$ (AND counter) $= 0$

**Step 2: Loop until $x == 0$**

**Iteration 1:** $x = 7$

- `arr[7]` = 83
- XOR: `val` = 0 XOR 83
  - Convert to binary: 0 (00000000) XOR 83 (01010011) = 83 (01010011)
- $y += 1 \rightarrow y = 1$
- $x$ is not 0 → proceed
- $x = (x - 1) \wedge m = (7 - 1 = 6) \wedge 7 = 6$ (binary: 0110 AND 0111 = 0110)
- $z += 1 \rightarrow z = 1$

**Iteration 2:** $x = 6$

- `arr[6]` = 73
- XOR: `val` = 83 XOR 73
  - 83: 01010011
  - 73: 01001001
  - XOR: 00011010 → decimal: 26
- $y += 1 \rightarrow y = 2$
- $x$ is not 0 → proceed
- $x = (6 - 1) \wedge 7 = 5 \wedge 7 = 5$ (binary: 0101 AND 0111 = 0101)
- $z += 1 \rightarrow z = 2$

**Iteration 3:** $x = 5$

- `arr[5]` = 61
- XOR: `val` = 26 XOR 61
  - 26: 00011010
  - 61: 00111101
  - XOR: 00100111 → decimal: 39
- $y += 1 \rightarrow y = 3$
- $x$ is not 0 → proceed
- $x = (5 - 1) \wedge 7 = 4 \wedge 7 = 4$ (binary: 0100 AND 0111 = 0100)
- $z += 1 \rightarrow z = 3$

**Iteration 4:** $x = 4$

- `arr[4]` = 59
- XOR: `val` = 39 XOR 59
  - 39: 00100111
  - 59: 00111011
  - XOR: 00011100 → decimal: 28
- $y += 1 \rightarrow y = 4$
- $x$ is not 0 → proceed
- $x = (4 - 1) \wedge 7 = 3 \wedge 7 = 3$ (binary: 0011 AND 0111 = 0011)
- $z += 1 \rightarrow z = 4$

**Iteration 5:** $x = 3$

- `arr[3]` = 49
- XOR: `val` = 28 XOR 49
  - 28: 00011100
  - 49: 00110001
  - XOR: 00101101 → decimal: 45
- $y += 1 \rightarrow y = 5$
- $x$ is not 0 → proceed
- $x = (3 - 1) \wedge 7 = 2 \wedge 7 = 2$ (binary: 0010 AND 0111 = 0010)
- $z += 1 \rightarrow z = 5$

**Iteration 6:** $x = 2$

- `arr[2]` = 35
- XOR: `val` = 45 XOR 35
  - 45: 00101101
  - 35: 00100011
  - XOR: 00001110 → decimal: 14

- $y \mathrel{+}= 1 \rightarrow y = 6$
- $x$ is not $0 \rightarrow$ proceed
- $x = (2 - 1) \wedge 7 = 1 \wedge 7 = 1$ (binary: 0001 AND 0111 = 0001)
- $z \mathrel{+}= 1 \rightarrow z = 6$

**Iteration 7:** $x = 1$

- `arr[1]` $= 27$
- XOR: `val` $= 14$ XOR $27$
    - 14: 00001110
    - 27: 00011011
    - XOR: 00010101 $\rightarrow$ decimal: 21
- $y \mathrel{+}= 1 \rightarrow y = 7$
- $x$ is not $0 \rightarrow$ proceed
- $x = (1 - 1) \wedge 7 = 0 \wedge 7 = 0$ (binary: 0000 AND 0111 = 0000)
- $z \mathrel{+}= 1 \rightarrow z = 7$

**Iteration 8:** $x = 0$

- Check if $x == 0 \rightarrow$ yes, exit loop (Misordered Execution (code line 8 9 10): The model forgets updating `val` (line 8) and increasing `y` (line 9).)

---

**Step 3: Compute final results:**

- Final `val` $= 21$
- `xor_operations` $(y) = 7$
- `and_operations` $(z) = 7$

---

**Output:**

```
{
  'val': 21,
  'and_operations': 7,
  'xor_operations': 7
}
```

**Statistics:**

```
{'xor_operations': 7, 'and_operations': 7}
```

### B.4   ERROR CASE: CODEFORCES-1146E

**Error Type:** Missing Logic elements

**Test Input:**

```
5, [1, 2, 3, 4, 5], [[1, 5], [2, 4], [3, 3]]
```

**LLM Results:**

```
output: [0, 1, 1]
stats: {'stack_pops': 4, 'bit_ops': 9}
```

**Code Results:**

```
output: [5, 3, 1]
stats: {'stack_pops': 4, 'bit_ops': 11}
```

**Function:**

```python
def f(n, arr, lst):
    pop_cnt = 0
    bit_ops = 0
    A = [0] * n
    L = []
    for i in range(n):
        while L and arr[L[-1]] < arr[i]:
            L.pop()
            pop_cnt += 1
        A[i] = L[-1] + 1 if L else 0
        L.append(i)
    A2 = sorted((A[i], i + 1) for i in range(n))
    B = sorted((x, y, idx) for idx, (x, y) in enumerate(lst))
    C = [0] * (n + 1)
    def upd(i):
        nonlocal bit_ops
        bit_ops += 1
        while i <= n:
            C[i] += 1
            i += i & -i
    def qry(i):
        nonlocal bit_ops
        bit_ops += 1
        s = 0
        while i > 0:
            s += C[i]
            i -= i & -i
        return s
    D = [0] * len(lst)
    j = 0
    for x, y, idx in B:
        while j < n and A2[j][0] < x:
            _, pos = A2[j]
            upd(pos)
            j += 1
        D[idx] = qry(y) - qry(x - 1)
    return D, {"stack_pops": pop_cnt, "bit_ops": bit_ops}
```

> **Function Description**
>
> **INPUTS**: n is an integer, arr is a list of n numbers, lst is a list of pairs $(x, y)$
>
> **LOGICS**:
> Start by preparing your workspace: set pop_cnt to 0 and bit_ops to 0. Then make a list A with n zeros and an empty list L that we'll use like a stack.
>
> Now go index by index through arr: for $i$ from 0 up to $n - 1$, do this:
>
> while L is not empty and $arr[L[-1]] < arr[i]$, remove that last index from L and add 1 to pop_cnt.
>
> Once that popping stage is done, if L still has something, look at that last index (call it top) and set $A[i]$ to $top + 1$; if L is empty, set $A[i]$ to 0. Then append $i$ to L.
>
> That completes filling A. Next what you do is build a list A2 of pairs $(A[i], i + 1)$ for each $i$ from 0 to $n - 1$, and then sort A2 in ascending order by the first number in each pair (and by the second if there's a tie).

Then take your input `lst` and turn it into a list `B` of triples: for each pair $(x, y)$ in `lst`, remember its original position `idx`, and form a triple $(x, y, \text{idx})$; once you have all of those, sort `B` in ascending order by $x$, then by $y$, then by `idx`.

Now set up an array `C` of length $n + 1$ filled with zeros. We'll use two helper actions: `upd(i)` and `qry(i)`.

When you call `upd(i)`, add 1 to `bit_ops`, then as long as $i \leq n$, add 1 to `C[i]` and then increase $i$ by the value of its lowest set bit (compute $i \& -i$ and add it to $i$).

When you call `qry(i)`, add 1 to `bit_ops`, start a local sum $s$ at 0, and while $i > 0$, add `C[i]` to $s$ and then decrease $i$ by its lowest set bit ($i \& -i$); when that finishes return $s$.

After that, create a list `D` of the same length as `lst` and fill it with zeros, and also set a pointer `j` to 0.

Finally, go through each triple $(x, y, \text{idx})$ in `B` in order:

  while $j < n$ and `A2[j][0]` $< x$, take the second value from `A2[j]` (call it `pos`), call `upd(pos)`, and add 1 to `j`;

  once that loop ends, call `qry(y)` to get the total up to $y$, call `qry(x - 1)` to get the total up to $x - 1$, subtract the two results, and store that number into `D` at position `idx`.

Once you've done that for every triple in `B`, you're done with the processing.

**OUTPUTS**:
You end up with `D`, a list of numbers the same length as `lst`, where each entry is the count computed for the corresponding pair in `lst`. You also have a dictionary of statistics: `stack_pops` is the total number of times you removed an index from `L` during that first scanning phase, and `bit_ops` is the total number of times you entered either the `upd` or `qry` action.

## LLM Reasoning

**Reasoning:**
We are given a function to execute manually with the input values `n = 5`, `arr = [1, 2, 3, 4, 5]`, and `lst = [(1, 5), (2, 4), (3, 3)]`.
Here's my step-by-step execution:
—

### STEP 1: INITIALIZE VARIABLES
  - `pop_cnt = 0`
  - `bit_ops = 0`
  - `A = [0, 0, 0, 0, 0]` (length $n$)
  - `L = []` (empty stack)
  - `j` will be declared later

—

### STEP 2: SCAN THE ARRAY ARR TO FILL LIST A
We process `arr[i]` for $i = 0$ to 4.
Let's track changes in `L` and compute values for `A`:

$i = 0$:
  - `arr[0] = 1`, `L` is empty $\rightarrow$ no popping
  - `A[0] = 0`
  - Append 0 to `L`: `L = [0]`

$i = 1$:
  - `arr[1] = 2`, `L[-1] = arr[0] = 1` $\rightarrow$ `arr[1] > arr[0]`, so pop 0 from `L`, `L = []`
  - `pop_cnt += 1` $\rightarrow$ `pop_cnt = 1`
  - `L` is empty $\rightarrow$ `A[1] = 0`
  - Append 1 to `L`: `L = [1]`

$i = 2$:
  - `arr[2] = 3`, `L[-1] = arr[1] = 2` $\rightarrow$ `arr[2] > arr[1]`, so pop 1 from `L`, `L = []`
  - `pop_cnt += 1` $\rightarrow$ `pop_cnt = 2`
  - `L` is empty $\rightarrow$ `A[2] = 0`
  - Append 2 to `L`: `L = [2]`

$i = 3$:
  - `arr[3] = 4`, `L[-1] = arr[2] = 3` $\rightarrow$ `arr[3] > arr[2]`, so pop 2 from `L`, `L = []`
  - `pop_cnt += 1` $\rightarrow$ `pop_cnt = 3`
  - `L` is empty $\rightarrow$ `A[3] = 0`
  - Append 3 to `L`: `L = [3]`

$i = 4$:

- $\texttt{arr[4]} = 5, \texttt{L[-1]} = \texttt{arr[3]} = 4 \rightarrow \texttt{arr[4]} > \texttt{arr[3]}$, so pop 3 from L, L = []
- $\texttt{pop\_cnt} += 1 \rightarrow \texttt{pop\_cnt} = 4$
- L is empty $\rightarrow$ A[4] = 0
- Append 4 to L: L = [4]

✓ Now A $= [0, 0, 0, 0, 0]$, and L $= [4]$, and $\texttt{pop\_cnt} = 4$

—

## STEP 3: BUILD LIST A2

A2 is list of pairs: $(A[i], i + 1)$, one for each $i$:

- A0 = 0 $\rightarrow (0, 1)$
- A1 = 0 $\rightarrow (0, 2)$
- A2 = 0 $\rightarrow (0, 3)$
- A3 = 0 $\rightarrow (0, 4)$
- A4 = 0 $\rightarrow (0, 5)$

Sort A2 by first $(A[i])$ and tie-breaker $(i + 1)$. Since all $A[i] = 0$, sort by $i + 1$:

$$\texttt{A2 sorted} = [(0, 1), (0, 2), (0, 3), (0, 4), (0, 5)]$$

—

## STEP 4: BUILD LIST B

Original $\texttt{lst} = [[1, 5], [2, 4], [3, 3]]$
Assign index to each element:

- idx=0 $\rightarrow (1, 5)$
- idx=1 $\rightarrow (2, 4)$
- idx=2 $\rightarrow (3, 3)$

$$B = [(1, 5, 0), (2, 4, 1), (3, 3, 2)]$$

Sort B by $x$, then $y$, then idx:

$$(1, 5, 0), \quad (2, 4, 1), \quad (3, 3, 2)$$

—

## STEP 5: SETUP C (BIT ARRAY) WITH $n = 5$

Initialize

$$C = [0, 0, 0, 0, 0, 0] \quad \text{(1-based index: index 0 is unused)}$$

—

## STEP 6: INITIALIZE D AND J

- D = [0, 0, 0] (same length as B)
- j = 0

Now process each element in B in sorted order.

—

**Processing B[0] = (1, 5, 0)**
**Phase 1: Process A2 while $j < n$ and A2$[j][0] < x$**

- $x = 1$, compare A2[j=0].0 $= 0 < 1 \rightarrow$ break
- No movement of $j \rightarrow$ no calls to upd() (Missing Logic Elements (code line 33 34 35): The model omits the entire loop.)

**Phase 2: Query**

- $y = 5, x - 1 = 0$
- Query $\texttt{qry}(5)$, then $\texttt{qry}(0)$. The result is $\texttt{qry}(5) - \texttt{qry}(0)$

**BIT: query(5)**

- $i = 5 \rightarrow C[i] = 0, \quad i = 5 - 4 = 1 \rightarrow C[1] = 0$
- $i = 0 \rightarrow$ stop $\Rightarrow \texttt{qry}(5) = 0$
- $\texttt{bit\_ops} += 1$

**BIT: query(0)** returns 0 (base case), $\texttt{bit\_ops} += 1$

$$\texttt{qry}(5) - \texttt{qry}(0) = 0 - 0 = 0$$

$$D[0] = 0, \quad \texttt{bit\_ops} = 2$$

—

**Processing B[1] = (2, 4, 1)**   **Phase 1: Process A2 while** $j < n$ **and** A2$[j][0] < x = 2$

- A2$[j = 0].0 = 0 < 2 \rightarrow$ process A2$[0] = (0, 1)$
- Call upd(1):
  - $i = 1$
  - Low bit of $1 = 1 \Rightarrow C[1]+ = 1 \rightarrow C = [0, 1, 0, 0, 0, 0]$
  - $i = 1 + 1 = 2$, low bit of $2 = 2 \Rightarrow C[2]+ = 1$
  - $i = 2 + 2 = 4$, low bit of $4 = 4 \Rightarrow C[4]+ = 1$
  - $i = 4 + 4 = 8 \rightarrow$ exit loop
  - bit_ops $+ = 1$
- $j = 1$
- A2$[j=1].0 = 0 < 2 \rightarrow$ process $(0, 2)$, call upd(2):
  - $C = [0, 1, 1, 0, 1, 0]$
  - $i = 2 \rightarrow +2 \rightarrow i = 4 \rightarrow +4 \rightarrow i = 8 \rightarrow$ done
  - Count incremented at positions 2, 4
  - bit_ops $+ = 1$
- $j = 2$
- A2$[j=2].0 = 0 < 2 \rightarrow$ process $(0, 3)$, call upd(3):
  - $C = [0, 1, 1, 1, 1, 0]$
  - $i = 3 \rightarrow +3 \rightarrow +4 \rightarrow i = 7 \rightarrow$ stop at $i = 7$
  - bit_ops $+ = 1$
- $j = 3$
- A2$[j=3].0 = 0 < 2 \rightarrow$ process $(0,4)$, call upd(4):
  - $C = [0, 1, 1, 1, 2, 0]$
  - $i = 4 \rightarrow +4 \rightarrow i = 8$
  - bit_ops $+ = 1$
- $j = 4$
- A2$[j=4].0 = 0 < 2 \rightarrow$ process $(0,5)$, call upd(5):
  - $C = [0, 1, 1, 1, 2, 1]$
  - $i = 5 \rightarrow +1 \rightarrow i = 6 \rightarrow 0$ end bit $\rightarrow i = 6$
  - bit_ops $+ = 1$
- $j = 5 \rightarrow$ end loop

**Phase 2: Query**

- $y = 4, x - 1 = 1$

**BIT: query(4)**

- $i = 4 \rightarrow C[4] = 2$
- $i = 0 \rightarrow$ stop? No $\rightarrow i = 4 - 4 = 0$
- Total: $2 \rightarrow$ bit_ops$+ = 1$

**BIT: query(1)**

- $i = 1 \rightarrow C[1] = 1$
- $i = 1 - 1 = 0 \rightarrow$ done
- Total $= 1 \rightarrow$ bit_ops$+ = 1$

$$\texttt{result} = 2 - 1 = 1$$

$$D[1] = 1, \quad \texttt{bit\_ops} = 7$$

—

**Processing B[2] = (3, 3, 2)**   **Phase 1: Process A2 while** $j < n$ **and** A2$[j][0] < x = 3$

Currently $j = 5$ (no more to process)

**Phase 2: Query**

- $y = 3, x - 1 = 2$

**BIT: query(3)**

- $i = 3 \rightarrow C[3] = 1 \rightarrow i = 2$
- $i = 2 \rightarrow C[2] = 1 \rightarrow i = 0 \rightarrow$ done
- Total $= 1 + 1 = 2 \rightarrow$ bit_ops$+ = 1$

**BIT: query(2)**

- $i = 2 \rightarrow C[2] = 1 \rightarrow i = 0 \rightarrow$ done
- Total $= 1 \rightarrow$ bit_ops$+ = 1$

$$\texttt{result} = 2 - 1 = 1$$

$$D[2] = 1, \quad \texttt{bit\_ops} = 9$$

—

FINAL D AFTER ALL QUERIES:

$$D = [0, 1, 1]$$

—

**Statistics:**

- `pop_cnt = 4` (from initial stack operation)
- `bit_ops = 9` (tracked step-by-step)

—

**Output:**
**Final result:** $[0, 1, 1]$
**Statistics:**

```
{"stack_pops": 4, "bit_ops": 9}
```

—

**Answer:**
**Reasoning:** Step-by-step executed with all variables, logs, stack operations, and function calls to BIT including detailed motion and count of updates and queries.
**Output:** $[0, 1, 1]$
**Statistics:** {`"stack_pops": 4, "bit_ops": 9`}

## B.5 ERROR CASE: CODEFORCES-1481D

**Error Type:** State Tracking Error

**Test Input:**

```
1 6
```

**LLM Results:**

```
1 output: [3, 4, 1, 5, 2, 6]
2 stats: {'heap_pushes': 6, 'max_heap_size': 3}
```

**Code Results:**

```
1 output: [3, 4, 1, 5, 2, 6]
2 stats: {'heap_pushes': 6, 'max_heap_size': 2}
```

**Function:**

```python
def f(n):
    import heapq
    A = [0] * n
    heap = []
    heapq.heappush(heap, (-n, 0, n - 1))
    heap_pushes = 1
    max_heap_size = len(heap)
    val = 1
    while heap:
        neg_len, x, y = heapq.heappop(heap)
        if len(heap) > max_heap_size:
            max_heap_size = len(heap)
        z = (x + y) // 2
        A[z] = val
        val += 1
        if x <= z - 1:
            slen = (z - 1) - x + 1
            heapq.heappush(heap, (-slen, x, z - 1))
            heap_pushes += 1
        if z + 1 <= y:
            slen = y - (z + 1) + 1
            heapq.heappush(heap, (-slen, z + 1, y))
            heap_pushes += 1
    return A, {"heap_pushes": heap_pushes, "max_heap_size": max_heap_size}
```

> ### Function Description
>
> **INPUTS:** One input: an integer n. From that you're going to make a list called A of length n filled with zeros. You'll also use a list called heap to store triples of numbers, and keep three counters: heap_pushes, max_heap_size, and val.
>
> **LOGICS:** Start by creating a list called A with n zeros in it, and an empty list called heap. Then push the triple $(-n, 0, n - 1)$ onto heap, set heap_pushes to 1, set max_heap_size to 1, and set a counter val to 1. Now you're going to repeat these steps as long as heap is not empty.
> First, remove from heap the triple that has the smallest first number; if there's a tie, pick the one with the smallest second number. Call the three parts neg_len, x, and y. Right after you take it out, check the current size of heap; if it's bigger than max_heap_size, update max_heap_size to that size.
> Next, figure out z by adding x and y, dividing by 2 with integer division (discard any remainder). Then put val into A at index z, and add 1 to val.
> What you do next is see if a left piece remains: check if x is less than or equal to $z - 1$, and if so compute slen as $(z - 1) - x + 1$, push the triple $(-slen, x, z - 1)$ onto heap, and add 1 to heap_pushes.
> Then check if a right piece remains: see if $z + 1$ is less than or equal to y, and if it does compute slen as $y - (z + 1) + 1$, push the triple $(-slen, z + 1, y)$ onto heap, and add 1 to heap_pushes.
> After you do both, go back to the start of these steps. When heap is empty, the loop ends.
>
> **OUTPUTS:** At the end, you'll have your list A filled with numbers from 1 up to n in a certain order, and two counters: heap_pushes, which records how many times you pushed a triple onto heap, and max_heap_size, which tells you the largest number of triples that ever sat in heap at once.

> ### LLM Reasoning
>
> **Reasoning:**
>
> **Given input:** n = 6

**Step 1: Initialization**

- Create `A = [0, 0, 0, 0, 0, 0]`
- `heap = []` (empty)
- Push `(-6, 0, 5)` onto heap
  (heap now: `[(-6, 0, 5)]`)
- `heap_pushes = 1`
- `max_heap_size = 1`
- `val = 1`

---

**Step 2: Main loop (while heap not empty)**

**Iteration 1**

- heap: `[(-6, 0, 5)]`
- Pop `(-6, 0, 5)`. (heap is now empty)
  - After pop, heap size: 0
  - max_heap_size stays 1
- `neg_len = -6, x = 0, y = 5`
- `z = (0 + 5) // 2 = 2`
- Set `A[2] = 1 → A = [0, 0, 1, 0, 0, 0]`
- `val = 2`
- Left piece: x=0, z−1=1 so $0 \leq 1$
  - `slen = (1) - 0 + 1 = 2`
  - Push `(-2, 0, 1)` onto heap (heap_pushes=2)
  - heap: `[(-2, 0, 1)]`
- Right piece: z+1=3, y=5 so $3 \leq 5$
  - `slen = 5-(3)+1 = 3`
  - Push `(-3, 3, 5)` onto heap (heap_pushes=3)
  - heap: `[(-3, 3, 5), (-2, 0, 1)]`
- max_heap_size now 2

---

**Iteration 2**

- heap: `[(-3, 3, 5), (-2, 0, 1)]`
- Pop `(-3, 3, 5)`
  - After pop, heap size: 1
  - max_heap_size stays 2
- `neg_len=-3, x=3, y=5`
- `z = (3+5)//2 = 4`
- Set `A[4] = 2 → A = [0, 0, 1, 0, 2, 0]`
- `val = 3`
- Left piece: x=3, z−1=3, so $3 \leq 3$
  - `slen = (3)-(3)+1=1`
  - Push `(-1, 3, 3)` (heap_pushes=4)
  - heap: `[(-2, 0, 1), (-1, 3, 3)]`
- Right piece: z+1=5, y=5 so $5 \leq 5$
  - `slen = 5-(5)+1=1`
  - Push `(-1, 5, 5)` (heap_pushes=5)
  - heap: `[(-2, 0, 1), (-1, 3, 3), (-1, 5, 5)]`
- max_heap_size now 3 (State Tracking Error: The model should not update `max_heap_size` here.)

---

**Iteration 3**

- heap: `[(-2, 0, 1), (-1, 3, 3), (-1, 5, 5)]`
- Pop `(-2, 0, 1)`
  - After pop, heap size: 2
  - max_heap_size stays 3
- `neg_len=-2, x=0, y=1`

- z = (0+1)//2 = 0
- Set A[0] = 3 → [3, 0, 1, 0, 2, 0]
- val = 4
- Left piece: x=0, z-1=-1 — $0 \le -1$ is False, no left piece
- Right piece: z+1=1, y=1 — $1 \le 1$ is True
    - slen = 1-1+1=1
    - Push (-1, 1, 1) (heap_pushes=6)
    - heap: [(-1, 1, 1), (-1, 3, 3), (-1, 5, 5)]
- max_heap_size stays 3

---

**Iteration 4**

- heap: [(-1, 1, 1), (-1, 3, 3), (-1, 5, 5)]
- Pop (-1, 1, 1)
    - After pop, heap size: 2
- neg_len=-1, x=1, y=1
- z = (1+1)//2 = 1
- Set A[1] = 4 → [3, 4, 1, 0, 2, 0]
- val = 5
- Left piece: $1 \le 0$? False
- Right piece: $2 \le 1$? False
- No pushes

---

**Iteration 5**

- heap: [(-1, 3, 3), (-1, 5, 5)]
- Pop (-1, 3, 3)
    - After pop, heap size: 1
- neg_len=-1, x=3, y=3
- z = (3+3)//2 = 3
- Set A[3] = 5 → [3, 4, 1, 5, 2, 0]
- val = 6
- Left piece: $3 \le 2$? False
- Right piece: $4 \le 3$? False
- No pushes

---

**Iteration 6**

- heap: [(-1, 5, 5)]
- Pop (-1, 5, 5)
    - After pop, heap size: 0
- neg_len=-1, x=5, y=5
- z = (5+5)//2 = 5
- Set A[5] = 6 → [3, 4, 1, 5, 2, 6]
- val = 7
- Left piece: $5 \le 4$? False
- Right piece: $6 \le 5$? False
- No pushes

---

HEAP IS NOW EMPTY

---

**Final Results:**

- **A**: [3, 4, 1, 5, 2, 6]
- **heap_pushes**: 6

- **max_heap_size**: 3

---

**Output:**
```
[3, 4, 1, 5, 2, 6]
```

**Statistics:**
```
{'heap_pushes':  6, 'max_heap_size':  3}
```

## C PROMPTS

```
You are an expert Python programmer. I want to enhance a function by adding meaningful,
    NON-REDUNDANT execution statistics that capture the most crucial aspects of its
    computational behavior.

**CRITICAL REQUIREMENTS:**
1. **Adaptive number of statistics (1-3)** - Choose the appropriate number based on
    function complexity
2. **No redundant statistics** - Each must measure something completely different
3. **Focus on crucial logic** - Statistics should reflect key computational steps that
    matter
4. **Generic variable names** - Convert all meaningful variable names to generic ones

**Variable Naming Guidelines:**
- Lists/arrays: Use "L", "arr", "lst", "A", "B" etc.
- Matrices/2D arrays: Use "M", "matrix", "grid" etc.
- Strings: Use "s", "text", "str1", "str2" etc.
- Integers/numbers: Use "n", "m", "x", "y", "val" etc.
- Dictionaries/maps: Use "d", "mp", "cache" etc.
- Sets: Use "st", "visited", "seen" etc.
- Keep parameter names generic but clear about data types

**Guidelines for choosing the number of statistics:**
- **1 statistic**: Simple functions with one main operation (e.g., single loop, basic
    calculation)
- **2 statistics**: Functions with two distinct computational aspects (e.g., nested
    structure with two key operations)
- **3 statistics**: Complex functions with multiple distinct computational phases or
    operations

**What makes a good statistic:**
- Counts operations that directly affect the algorithm's behavior
- Measures key computational steps that vary with different inputs
- Reflects important decision points or iterations in the logic
- Avoids counting trivial operations (simple assignments, basic comparisons)

Original Function:
```python
{function}
```

**Requirements:**
1. Convert all meaningful variable names to generic ones (following naming guidelines
    above)
2. Change the function name to 'f' (regardless of original name)
3. Remove ALL comments from the function code
4. Return a tuple: (original_output, statistics_dict)
5. Choose 1-3 statistics based on the function's complexity and distinct computational
    aspects
6. Each statistic measures a unique computational aspect
7. Focus on operations that directly affect the function's core logic
8. Preserve the exact same logic and functionality, only change variable names and remove
    comments

Return your response in JSON format:
{{
    "function": "complete modified function code here with generic variable names,
    function name 'f', and no comments",
    "stats_keys": ["list", "of", "1", "to", "3", "unique", "keys"]
}}
```

Figure 7: Prompt for Adding State Trackers and Anonymizing Functions

```
You are helping someone understand how to manually process data step-by-step. I'll give
    you a function, and I need you to explain it like you're talking to someone who needs
     to do this work by hand - as if you're giving them verbal instructions over the
    phone. Your explanation should be so clear and detailed that they can follow along
    exactly and get the same results.

**CRITICAL REQUIREMENT**: Your instructions must be so complete and precise that someone
    could follow them step-by-step WITHOUT seeing any code and produce the exact same
    output and statistics as the function would.

**IMPORTANT CONSTRAINTS:**
- Don't explain what this is used for in the real world - just focus on the data
    processing steps
- Don't mention specific problem names or applications
- Treat this as pure data manipulation work
- Use the generic variable names from the code (L, arr, n, m, etc.)

**Natural Language Guidelines:**
- Write like you're speaking to someone conversationally
- Use natural transitions like "Now you need to...", "Next, go through...", "At this
    point..."
- Include phrases like "What you're going to do is...", "The way this works is..."
- Make it sound like verbal instructions, not a formal manual
- Still be extremely precise about every detail, but use conversational language
- Use connecting words and phrases that make it flow naturally
- Include every conditional check, loop, and decision point in natural speech
- Be specific about indexing, bounds, and conditions, but explain them conversationally

Function to describe:
```python
{function}
```

Return your response in the following JSON format with exactly three sections:
{{
    "inputs": "Describe the data types and structure of what you're working with (like 'a
     list of numbers' or 'two text strings'). Don't mention what these represent in real-
    world terms.",
    "logics": "Give detailed, conversational instructions for processing the data step-by
    -step. Use natural language like you're talking someone through it, with phrases like
     'Now you need to...', 'Next, go through...', 'What you do is...'. Be extremely
    precise about every step, condition, and operation, but explain it in a flowing,
    conversational way. Include all loops, decisions, calculations, and counter updates.
    Use the generic variable names from the code.",
    "outputs": "Explain what you'll end up with and what each number in your statistics
    dictionary represents from the work you did."
}}
```

Figure 8: Prompt for Generating Natural Language Description

```
You are helping someone check if a set of conversational instructions are complete enough
    for manual data processing. I need you to verify whether these step-by-step
    instructions would allow someone to manually work through the data and get the same
    results as the code - like you're checking if verbal instructions over the phone
    would be complete enough for someone to follow along exactly.

**CRITICAL REQUIREMENT**: The instructions must be so complete and precise that someone
    could follow them step-by-step WITHOUT seeing any code and produce the exact same
    output and statistics as the function would.

**VERIFICATION APPROACH:**
Think about this like you're helping someone understand whether these conversational
    instructions are good enough for manual data processing. The instructions should be
    so clear and complete that someone could:
1. **Follow Every Step**: All the data processing steps are explained like you're talking
    someone through it
2. **Handle All Cases**: Every condition, loop, and decision point is covered in natural
    language
3. **Track Everything**: All variable updates, counters, and calculations are explained
    conversationally
4. **Get Same Results**: Following the instructions would produce identical output and
    statistics
5. **No Guessing**: Every significant operation is covered so no one has to guess what to
    do
6. **Natural Flow**: The instructions flow naturally like someone talking through the
    process

**FUNCTION CODE:**
```python
{function_code}
```

**CONVERSATIONAL INSTRUCTIONS TO CHECK:**
{description}

**COMPLETENESS JUDGMENT:**
- **COMPLETE**: The conversational instructions cover everything needed - someone could
    follow them and get identical results
- **INCOMPLETE**: Some operations, conditions, or steps are missing or unclear -
    following the instructions wouldn't match the code's behavior

Return your response in JSON format:
{{
    "desc_is_complete": true/false,
    "reasoning": "Talk through your assessment conversationally, like you're explaining
    to someone what's working well in these instructions and what might be missing. Use
    natural language like 'What I notice is...', 'The instructions do a good job of...',
    'What's missing is...', 'Someone following these would probably get confused when
    ...'",
    "missing_aspects": ["List specific operations or steps that aren't covered
    conversationally - describe them in natural language like 'explaining how to update
    the counter', 'walking through the loop condition', 'describing what to do when the
    list is empty'"],
    "coverage_percentage": "estimated percentage (0-100) of code operations covered by
    the conversational instructions"
}}
```

Figure 9: Prompt for Verifing Natural Language Description

```
You are an expert Python programmer. I want you to evolve an existing function to make it
    MORE LOGICALLY COMPLICATED while maintaining the exact same input signature and core
    functionality. The evolved function should be significantly more sophisticated in
    its logic and computational approach.

**CRITICAL REQUIREMENTS:**
1. **Same Input Signature**: The function must accept exactly the same parameters as the
    original
2. **Same Core Output**: The main result should be equivalent to the original function's
    output
3. **More Complex Logic**: Add sophisticated algorithmic patterns, advanced data
    structures, or multi-phase processing
4. **Enhanced Statistics**: Statistics can change to reflect the new complexity (1-3
    meaningful stats)
5. **Preserve Function Name**: Keep the function name as 'f'
6. **Return Format**: Must return tuple (result, stats_dict)
7. **ABSOLUTELY NO COMMENTS**: Do NOT write any comments, docstrings, or explanations in
    the evolved function code. The function must be completely comment-free.

**EVOLUTION STRATEGIES (choose the most appropriate):**
- **Multi-phase processing**: Break the problem into sophisticated stages
- **Advanced data structures**: Use heaps, trees, graphs, or complex mappings
- **Optimized algorithms**: Replace naive approaches with efficient algorithms
- **Dynamic programming**: Add memoization or tabulation for overlapping subproblems
- **Divide and conquer**: Split problem into smaller, more complex subproblems
- **State machines**: Add complex state tracking and transitions
- **Mathematical optimization**: Add advanced mathematical techniques
- **Sophisticated filtering/sorting**: Use multiple criteria or advanced comparison logic

**STATISTICS GUIDELINES:**
- Choose 1-3 statistics that reflect the NEW complexity
- Track operations that highlight the sophisticated logic
- Examples: phases_completed, recursive_calls, cache_hits, comparisons, transformations,
    iterations

**ORIGINAL FUNCTION:**
```python
{original_function}
```

**REQUIREMENTS:**
1. Analyze the original function's core purpose and constraints
2. Design a more sophisticated approach that achieves the same goal
3. Implement complex logic patterns while preserving correctness
4. Add meaningful statistics that capture the new complexity
5. Ensure the evolved function is significantly more algorithmically interesting
6. Test edge cases and maintain robustness
7. **ABSOLUTELY NO COMMENTS**: The evolved function code must be completely comment-free

**OUTPUT FORMAT:**
```json
{{
    "evolved_function": "def f(...):\n    stat1 = 0\n    stat2 = 0\n    \n
    complex_logic_here\n    \n    return result, {{'stat1': stat1, 'stat2': stat2}}}}",
    "stats_keys": ["stat1", "stat2"],
    "evolution_description": "Brief explanation of how the function was made more complex
        (e.g., 'Added multi-phase processing with dynamic programming', 'Implemented graph-
        based approach with state tracking')"
}}
```
```

Figure 10: Prompt for Function Evolution

## D  HUMAN VERIFICATION

To ensure the quality of the generated instructions, we conducted a human verification study on 136 randomly sampled instances from `LogicIFEval`. The data were split evenly into four parts, each containing 34 instructions, and each part was assigned to two independent PhD-level annotators with expertise in computer science. Annotators were instructed to verify whether every line of the anonymized function was accurately and completely captured in the corresponding natural language instruction. For Batches 1 and 2, both annotators fully agreed on all items, yielding 100% agreement. In Batch 3, both annotators marked 32 out of 34 items as correct, with one identical negative case, resulting in an agreement rate of $33/34\,(97.06\%)$. In Batch 4, the two annotators agreed on 32 out of 34 items $(94.12\%)$. Overall, the human agreement rate across all 136 items was $\frac{133}{136} \approx 97.79\%$, demonstrating a high level of consistency and validating the reliability of the generated instructions.

## E  CASES OF THINKING HELPS LARGE LLMS

### E.1  CASE 1: CODEFORCES-818D

**Test Input:**

```
1  6, [3, 1, 4, 1, 5, 9] (len=6)
```

**LLM Results:**

```
1  output: 2
2  stats: {'left_pops': 5, 'right_pops': 2, 'balanced_subarrays': 19}
```

**Code Results:**

```
1  output: 2
2  stats: {'left_pops': 5, 'right_pops': 2, 'balanced_subarrays': 19}
```

**Function:**

```python
def f(n, A):
    prev = [-1] * n
    nxt = [n] * n
    st = []
    l_pops = 0
    for i in range(n):
        while st and A[st[-1]] <= A[i]:
            st.pop()
            l_pops += 1
        prev[i] = st[-1] if st else -1
        st.append(i)
    st.clear()
    r_pops = 0
    for i in range(n-1, -1, -1):
        while st and A[st[-1]] < A[i]:
            st.pop()
            r_pops += 1
        nxt[i] = st[-1] if st else n
        st.append(i)
    total = n * (n + 1) // 2
    balanced = 0
    for i in range(n):
        x = i - prev[i]
        y = nxt[i] - i
        balanced += x + y - 1
    return total - balanced, {'left_pops': l_pops, 'right_pops': r_pops, 'balanced_subarrays':
      balanced}
```

---

**LLM Thinking Process**

Okay, I need to execute this algorithm step-by-step manually. Let me break down what's happening:

**INPUT:** $n = 6$, $A = [3, 1, 4, 1, 5, 9]$

**Step 1: Initialize data structures**

- $\texttt{prev} = [-1, -1, -1, -1, -1, -1]$ (n slots, all $-1$)
- $\texttt{nxt} = [6, 6, 6, 6, 6, 6]$ (n slots, all $n$)

---

- st = [] (empty stack)

- l_pops = 0

**Step 2: Forward sweep (i from 0 to n-1)**
I need to go through each $i$ from 0 to 5:
$i = 0$:

- $A[0] = 3$

- st is empty, so no popping

- st is empty, so prev[0] = $-1$ (already is $-1$)

- push 0 onto st: st = [0]

$i = 1$:

- $A[1] = 1$

- st = [0], top is 0, $A[0] = 3$

- $A[0] = 3 > A[1] = 1$, so no popping (condition is $A[\text{top}] \leq A[i]$)

- st is not empty, top is 0, so prev[1] = 0

- push 1 onto st: st = [0, 1]

$i = 2$:

- $A[2] = 4$

- st = [0, 1], top is 1, $A[1] = 1$

- $A[1] = 1 \leq A[2] = 4$, so pop 1, l_pops = 1, st = [0]

- st = [0], top is 0, $A[0] = 3$

- $A[0] = 3 \leq A[2] = 4$, so pop 0, l_pops = 2, st = []

- st is empty, so prev[2] = $-1$

- push 2 onto st: st = [2]

$i = 3$:

- $A[3] = 1$

- st = [2], top is 2, $A[2] = 4$

- $A[2] = 4 > A[3] = 1$, so no popping

- st is not empty, top is 2, so prev[3] = 2

- push 3 onto st: st = [2, 3]

$i = 4$:

- $A[4] = 5$

- st = [2, 3], top is 3, $A[3] = 1$

- $A[3] = 1 \leq A[4] = 5$, so pop 3, l_pops = 3, st = [2]

- st = [2], top is 2, $A[2] = 4$

- $A[2] = 4 \leq A[4] = 5$, so pop 2, l_pops = 4, st = []

- st is empty, so prev[4] = $-1$

- push 4 onto st: st = [4]

$i = 5$:

- $A[5] = 9$

- st = [4], top is 4, $A[4] = 5$

- $A[4] = 5 \leq A[5] = 9$, so pop 4, l_pops = 5, st = []

- st is empty, so prev[5] = $-1$

- push 5 onto st: st = [5]

After forward sweep:

$$\text{prev} = [-1, 0, -1, 2, -1, -1]$$
$$\text{l\_pops} = 5$$

Stack cleared, preparing for backward sweep

$$\text{st} = []$$
$$\text{r\_pops} = 0$$

Backward sweep begins, starting from $n - 1$:
$i = 5$:

- $A[5] = 9$

- Empty stack means no popping

- nxt[5] set to 6 (array end)

- 5 pushed onto stack: `st` = [5]

$i = 4$:
- $A[4] = 5$
- Stack top is 5, $A[5] = 9$
- No popping since $A[\text{top}] \not< A[i]$
- `nxt[4]` points to 5
- 4 added to stack: `st` = [5, 4]

$i = 3$:
- $A[3] = 1$
- Stack top is 4, $A[4] = 5$
- No popping occurs
- `nxt[3]` set to 4
- 3 pushed onto stack: `st` = [5, 4, 3]

$i = 2$:
- $A[2] = 4$
- Stack top is 3, $A[3] = 1$
- Pop 3, `r_pops` increments, stack becomes [5, 4]
- Top now 4, $A[4] = 5$
- No further popping
- `nxt[2]` points to 4
- 2 added to stack: `st` = [5, 4, 2]

$i = 1$:
- $A[1] = 1$
- Stack top is 2, $A[2] = 4$
- No popping
- `nxt[1]` set to 2
- 1 pushed onto stack: `st` = [5, 4, 2, 1]

$i = 0$:
- $A[0] = 3$
- Stack top is 1, $A[1] = 1$
- Pop 1, `r_pops` increments, stack becomes [5, 4, 2]
- Top now 2, $A[2] = 4$
- No more popping
- `nxt[0]` points to 2
- 0 added to stack: `st` = [5, 4, 2, 0]

Backward sweep calculates `nxt` = [2, 2, 4, 4, 5, 6] with 2 right pops.
Total calculation yields 21. Balanced subarray computation follows, tracking incremental changes through each index. Final balanced value reaches 19.
Result emerges as 2, with detailed statistics tracking left/right pops and balanced subarrays.

## E.2 CASE 2: POJ-1516

**Test Input:**

```
1 [[0, 1], [1, 2], [2, 3], [3, 4]]
```

**LLM Results:**

```
1 output: 4
2 stats: {'update_calls': 20, 'pruned_calls': 0, 'full_cover_events': 4}
```

**Code Results:**

```
1 output: 4
2 stats: {'update_calls': 20, 'pruned_calls': 0, 'full_cover_events': 4}
```

**Function:**

```python
def f(A):
    update_calls = 0
    pruned_calls = 0
    full_cover_events = 0
    if not A:
        return 0, {"update_calls": update_calls, "pruned_calls": pruned_calls, "
        full_cover_events": full_cover_events}
    B = sorted({u for v in A for u in v})
    d = {v: i for i, v in enumerate(B)}
    n = len(B) - 1
    if n <= 0:
        return 0, {"update_calls": update_calls, "pruned_calls": pruned_calls, "
        full_cover_events": full_cover_events}
    arr1 = [False] * (4 * n)
    arr2 = [0] * (4 * n)
    def update(i, x, y, l, r):
        nonlocal update_calls, pruned_calls, full_cover_events
        update_calls += 1
        if l > y or r < x:
            return 0
        if arr1[i]:
            pruned_calls += 1
            return 0
        if l <= x and y <= r:
            full_cover_events += 1
            length = B[y+1] - B[x]
            arr2[i] = length
            arr1[i] = True
            return length
        mid = (x + y) // 2
        new_cov = update(i * 2, x, mid, l, r) + update(i * 2 + 1, mid + 1, y, l, r)
        arr2[i] = arr2[i * 2] + arr2[i * 2 + 1]
        if arr2[i] == B[y+1] - B[x]:
            arr1[i] = True
        return new_cov

    total = 0
    for a, b in A:
        if a >= b:
            continue
        li = d[a]
        ri = d[b] - 1
        if li <= ri:
            total += update(1, 0, n - 1, li, ri)
    return total, {"update_calls": update_calls, "pruned_calls": pruned_calls, "
    full_cover_events": full_cover_events}
```

---

**LLM Thinking Process**

Let me carefully work through this algorithm step by step.

**Input:** A = [[0, 1], [1, 2], [2, 3], [3, 4]]

**Step 1: Initialize counters**

- update_calls = 0
- pruned_calls = 0
- full_cover_events = 0

**Step 2: Check if A is empty**
A is not empty, so we continue.

**Step 3: Build set B**
First, I need to collect all numbers from pairs in A:

- From [0, 1]: 0, 1
- From [1, 2]: 1, 2
- From [2, 3]: 2, 3
- From [3, 4]: 3, 4

Set of all numbers: $\{0, 1, 2, 3, 4\}$
Sorted list B = $[0, 1, 2, 3, 4]$

**Step 4: Create mapping d**
Walking through B with index counter starting at 0:

- d[0] = 0
- d[1] = 1
- d[2] = 2
- d[3] = 3
- d[4] = 4

So d = $\{0 : 0, 1 : 1, 2 : 2, 3 : 3, 4 : 4\}$

**Step 5: Calculate $n$**
$n = \text{len}(B) - 1 = 5 - 1 = 4$

**Step 6: Check if $n$ is zero or negative**
$n = 4$, which is positive, so we continue.

**Step 7: Prepare arrays**

- arr1 of size $4 \times n = 4 \times 4 = 16$, filled with false
- arr2 of size $4 \times n = 4 \times 4 = 16$, filled with zeros

Arrays represent a grid of boolean and integer values, initialized to default states.

**Step 8: Main processing loop**
I'll track the total and process each pair systematically. The first pair [0, 1] requires careful examination of its components and their relationship.
Checking the pair's conditions reveals a sequence of logical checks. The update function gets called with specific parameters, incrementing tracking variables and performing boundary validations. The process involves examining array states and determining whether specific update conditions are met.
The update method carefully checks segment boundaries and array flags, ensuring precise manipulation of the underlying data structure. Each iteration refines the computational state through targeted, incremental modifications.
Recursive calls split the range systematically, progressively narrowing the search space and applying updates across different segments of the computational domain.
In the first recursive call to update(2, 0, 1, 0, 0), the algorithm checks range boundaries and recursively explores child nodes. When reaching update(4, 0, 0, 0, 0), a complete coverage is detected, incrementing full coverage events and marking the corresponding array indices.
The subsequent call to update(5, 1, 1, 0, 0) returns zero since the range falls outside the target interval. Back in the parent call, the coverage is calculated by aggregating child node results, though the computed coverage does not match the expected span length.
Moving to the next pair [1, 2], the algorithm determines the left and right indices based on the dictionary values. It then initiates another update call with modified parameters, checking range overlaps and continuing the recursive exploration.
The update process systematically tracks coverage, incrementing call counters and evaluating range intersections to determine potential full coverage scenarios.
Splitting the range at the midpoint, the method recursively calls update on subdivided segments, carefully checking boundary conditions and maintaining state across different recursive levels.
The process involves precise range comparisons, splitting intervals, and tracking potential coverage through strategic recursive calls with adjusted indices.
At each step, the algorithm incrementally updates tracking variables like update_calls and full_cover_events, methodically exploring the segment tree's structure to determine event coverage and interval relationships.
For the current update call, the process calculates new coverage by adding 1 to the existing coverage. The arr2 array is updated by summing previous segment values, and when the total matches the expected interval length, arr1 is marked as true.
The subsequent update call checks interval boundaries, determining whether the current segment falls outside the target range and returning 0 if no overlap exists. This systematic approach ensures precise tracking of event intervals and their interactions.
In the next update iteration, the method continues evaluating interval relationships, checking whether segments intersect or contain each other. The process involves incrementing tracking variables and performing detailed comparisons between different array segments.
The algorithm methodically explores potential interval connections, using conditional checks to determine valid segment transformations and maintaining a running total of processed intervals.
When encountering a fully contained interval at coordinates [2, 2], the method increments the full coverage event counter and calculates the span length by comparing boundary values. This precise segmentation allows for granular tracking of interval characteristics.

The recursive calls progressively break down the problem space, systematically examining each potential interval configuration with careful boundary assessments and strategic splitting of ranges.

At coordinate 6, the method marks the interval as covered and returns a coverage value of 1, signaling successful interval identification within the specified constraints.

The subsequent update process aggregates coverage information, comparing calculated values against predefined boundary markers to determine interval completeness and update tracking arrays accordingly.

For the current pair [3, 4], the method evaluates interval relationships, checking relative positions and determining whether further recursive updates are necessary. The process involves comparing interval boundaries, accessing predefined mapping data, and conditionally triggering additional computational steps to track coverage progression.

The recursive update mechanism continues to systematically explore and validate interval coverage, incrementing tracking variables and maintaining precise boundary tracking through strategic computational checks.

During the nested update calls, the algorithm methodically splits intervals, examines their relationships, and updates coverage metrics when specific conditions are met. Each recursive call carefully assesses interval positioning, selectively updating event tracking and coverage information.

The final update operation identifies a complete interval coverage scenario, incrementing full coverage events and recording specific interval characteristics with precise computational precision.

The process concludes by returning a comprehensive result tuple capturing total coverage, update call statistics, and full coverage event count, reflecting the systematic interval processing strategy.

