# OpenReview forum: "Complex Logical Instruction Generation"
_ICLR.cc/2026/Conference — Submitted to ICLR 2026_

### Official Review · Reviewer_xW3R · 2025-10-28

**Soundness:** 3
**Presentation:** 3
**Contribution:** 2
**Rating:** 4
**Confidence:** 4

**Summary:**

This paper proposes LogicIFGen, a framework which enables automatic generation of test instances for logic relations based on code. The framework is scalable and verifiable. Using this framework, the authors construct LogicIFEval, a benchmark curated to evaluate LLM capabilities in logic understanding and instruction following. Extensive empirical experiment results show that many of the models have poor performance on the benchmark, highlighting the issues in using LLMs to handle complex logic queries.

**Strengths:**

1. The proposed framework, LogicIFGen, is scalable and verifiable. The authors also conduct human studies to show that the generation is of high quality.
2. The research question itself is interesting and important: LLMs have to be able to follow various types of instructions, which potentially involve complex logic relations, to properly serve users.
3. Writing and presentation are very clear. It is very easy to understand the authors' points.

**Weaknesses:**

1. Related work section is not comprehensive enough. It reviews many works which are relevant but not closely relevant to the research question in the general LLM reasoning area. More strongly related works should be thoroughly discussed as I will elaborate below.

2. The novelty of the research question is not extremely clear given many existing works in highly similar areas. This is my biggest concern. I think there are many existing works in code execution which are highly relevant, but not fully discussed in the related work section. For example, [1,2,3,4, inter alia] have done comprehensive analysis on the limitations of LLMs in simulating code execution result, trace, and executing natural language instructions which can be solved/verified by code. The authors should more carefully review relevant literature and argue for their novelty.

[1] La Malfa, E., Weinhuber, C., Torre, O., Lin, F., Marro, S., Cohn, A., ... & Wooldridge, M. (2024). Code simulation challenges for large language models. arXiv preprint arXiv:2401.09074.

[2] La Malfa, E., Weinhuber, C., Torre, O., Lin, F., Huang, X. A., Marro, S., ... & Wooldridge, M. (2025). Code Simulation as a Proxy for High-order Tasks in Large Language Models. arXiv preprint arXiv:2502.03568.

[3] Sun, S., Hsieh, C. P., Ladhak, F., Arakelyan, E., Serano, S. A., & Ginsburg, B. (2025). L0-Reasoning Bench: Evaluating Procedural Correctness in Language Models via Simple Program Execution. arXiv preprint arXiv:2503.22832.

[4] Liu, C., Dylan Zhang, S., Ibrahimzada, A. R., & Jabbarvand, R. (2024). Codemind: A framework to challenge large language models for code reasoning. arXiv e-prints, arXiv-2402.

**Questions:**

1. It would be really helpful if the authors could kindly address concerns mentioned above.

2. The theme of this paper is testing complex logic. What is the great advantage of using code as backbone to generating the questions instead of using classic formal logic frameworks such as first-order logic? The formal logic is also verifiable and easily programmable.

---

> ### Author Response · Authors · 2025-11-15
> **Response to Weakness 1: Related Work of LLM Reasoning**
>
> Thanks for the comment. Our work primarily focuses on **instruction following**, rather than **reasoning**. First, the instructions we generate explicitly tell the model what to do at each step, in contrast to typical reasoning tasks (e.g., math reasoning) where the model must infer what to do next. Second, we perform rigorous data filtering to ensure that **each step of the instruction is simple enough for the model to execute directly, without requiring additional external reasoning effort** (see Section 3).
>
> That said, we do observe in our experiments that models with explicit “thinking” mechanisms achieve higher accuracy in our evaluation (see Table 1 and Section 5.2). And **we have discussed the works on the relationship between reasoning and instruction following in the Section Related Work (lines 495-499)**.

---

> ### Author Response · Authors · 2025-11-15
> **Response to Weakness 2: Difference with Code Execution**
>
> Thanks for the valuable comment and pointing out these related works! Although they appear similar at first glance, they differ from our approach in several fundamental ways:
> - The line of work on code execution or code simulation tasks primarily takes code snippets as input and asks LLMs to act as interpreters by predicting the execution results (e.g., CRUXEval https://arxiv.org/pdf/2401.03065, LiveCodeBench https://arxiv.org/abs/2403.07974, L0-Reasoning Bench https://arxiv.org/pdf/2503.22832v2). In contrast, the high-level goal of our work is to evaluate LLMs’ ability to follow complex logical instructions expressed in natural language. We use code functions only as a source to synthesize natural language instructions that contain rich logical structures, and then test whether LLMs can faithfully follow these instructions. Thus, the input to our task is **natural language**, not code. **Understanding the logical structures embedded in natural language is a more general and practically important capability**, as today’s agentic workflows and operational constraints are predominantly described using natural language. A notable example is the Claude system instructions (https://www.dbreunig.com/2025/05/07/claude-s-system-prompt-chatbots-are-more-than-just-models.html), which could contain description of complicated agentic model behaviour evolving condition, error handling, or function calling. Evaluating how well LLMs can follow such instructions is precisely the problem our work aims to address.
> - We would like to clarify the relationship between our work and the works you mentioned that appear similar.
>     - **First**, regarding https://arxiv.org/pdf/2502.03568: This work uses code execution as a proxy for evaluating the reasoning ability of LLMs. It converts natural language reasoning tasks into code and shows a strong correlation between a model’s ability to solve the task in either code or natural language. While this direction is relevant, the motivation is fundamentally different from ours. Their goal is to test reasoning by translating problems into executable programs. In contrast, our goal is to evaluate instruction-following ability, and we design a complete data-synthesis framework and curate diverse code sources to generate logic-rich natural language instructions. We do not evaluate code execution; instead, we test whether LLMs can follow logically structured instructions expressed in natural language.
>     - **Second**, our work shares one high-level motivation with L0-Reasoning Bench (https://arxiv.org/pdf/2503.22832v2): Can LLMs reliably execute simple rules across multiple incremental steps? Both works use programs as sources to construct evaluation data. However, the data format and synthesis procedures are fundamentally different. L0-Reasoning Bench uses simple code snippets and evaluates whether models can correctly predict execution results at each step. Their input to the model is code. In contrast, our benchmark is fully in natural language form. The evaluation instances in our evaluation data not only express step-by-step rules but also incorporate complex logical structures to chain the rules.
>
> We have incorporated these discussions into the Related Work of the **updated version** of the paper (in blue). Thanks again for highlighting them!

---

> ### Author Response · Authors · 2025-11-15
> **Response to Question: Code Rather than Formal Logic**
>
> Thanks for the question! While formal logic provides a declarative way to express symbolic relationships, it does not naturally encode many of the logical patterns that appear in real natural-language instructions, such as ordered execution, function calling, or conditional branching. These logical structures are difficult to represent compactly in formal logic without moving to more complex systems, but they are expressed transparently and natively in code.
>
> Moreover, our experiments (see Table 1 in the paper) show that the logical complexity of code functions (e.g., number of control flow, nesting depth) could be well used to measure the logical difficulty of natural language instructions, which is a great feature for generating instructions of different difficulty levels.
>
> Finally, code offers strong scalability: given any executable function and test cases, our proposed framework LogicIFGen can automatically synthesize corresponding natural-language instructions, enabling large-scale, diverse, and verifiable data generation for training and evaluation.

---

> ### Author Response · Authors · 2025-11-21
>
> Dear Reviewer xW3R,
>
> We have posted our responses to your questions and concerns. When you have time, please take a look. We hope our clarifications address your points. If you have any further questions or would like additional discussion, feel free to let us know.
>
> Best regards

---

> ### Author Response · Authors · 2025-11-26
>
> Dear Reviewer xW3R,
>
> Thank you for your time and for providing such thorough and high-quality reviews of our paper. We sincerely appreciate the thoughtful feedback. We have carefully addressed each of your concerns in our response and have also provided an updated version of the paper reflecting the revisions.
>
> Please feel free to take a look at your convenience. If any of our responses remain unclear or if further questions arise, we would be very happy to engage in additional discussion.
>
> Best regards,
>
> The Authors

---

### Official Review · Reviewer_MrYZ · 2025-10-29

**Soundness:** 2
**Presentation:** 2
**Contribution:** 1
**Rating:** 2
**Confidence:** 3

**Summary:**

The paper introduces LogicIFGen, a method that uses an LLM to anonymize functional programs. This method further introduces state tracking variables and translates the programs into a natural language description of the function. The method is used to create a benchmark dataset, called LogicIFEval, for evaluating the ability of LLMs to follow instructions when asked to execute the provided natural language instruction step-by-step for several test cases. The complexity of the instructions is captured using a syntax-tree-based measure on the anonymized programs. Several frontier LLMs were tested on the benchmark, revealing declining performance with increasing instruction complexity. Further analysis of the models' different failure modes was conducted.

**Strengths:**

Overall, the paper is easy to follow. The experiments appear to be fully documented and reproducible, and the topic of assessing LLM abilities in relation to the logical complexity of the task is highly relevant.

**Weaknesses:**

1.  The paper's main weakness is the limited relevance and coherence of the analysis conducted. Although assessing the abilities of LLM in instruction-following tasks by investigating dependency on increasing levels of logical complexity is promising, the main results focus on comparing the performance of different models. The overall pattern of declining performance with increasing instruction complexity is briefly discussed. However, the subsequent analysis of different failure modes does not consider varying difficulty levels. Hence, there is no insight into whether or how these failure modes depend on instruction complexity. This does not align with the paper's motivation and potential novelty. To significantly improve the paper's quality, the benchmark and related analysis should focus on the connections between LLM behavior and the logical complexity of instructions. This would allow the paper to fulfill the promises made in the abstract and introduction.
2.  A second weakness is the lack of discussion or theory on the motivation behind the introduced dataset. Why is it important to explicitly prohibit the model from using programming or tools in general? Does the benchmark provide evidence for or against systematic compositionality in LLM behavior?

In summary, the paper appears to be immature in its current state. It presents an interesting idea for using LLM to process a selection of functions into datasets for assessing LLM instruction-following abilities in relation to task complexity. However, it lacks a sufficient theory explaining why and how these abilities can be assessed with the proposed dataset. Additionally, the analysis is insufficient for meaningfully evaluating the LLM behavior related to task complexity.

**Questions:**

1.  Table 1 should specify whether performance is a percentage of test cases or questions. The number of questions in brackets after the difficulty levels seems to indicate a percentage of questions. However, it is unclear whether a question is counted if only some test cases are correct in terms of output, state, or both. The numbers appear to be a percentage of test cases, especially when compared to Figure 3. For example, GPT-5 has state tracking errors in 28.1% of questions but could not achieve a "state" performance of 89.44% correct answers; rather, it achieved this result for only a subset of test cases.
2.  Why is the fourth result group labeled "Average" when the subtitle says "Overall"? The most intuitive average would be the weighted and unweighted means of the three columns on the left, respectively. However, that is not the case. Is this about overall performance rather than an average?
3.  Figure 3 needs a subtitle and should specify if it is about at least one of the specific failure modes in some test case per question.

---

> ### Author Response · Authors · 2025-11-15
> **Response to Weakness 1: More Detailed Results of the Relationship between Logical Complexity and LLM Behavior**
>
> Thanks for the valuable comment. Below we provide a more fine-grained breakdown of model performance across different instruction-difficulty levels. Specifically, we partition the 426 tasks in LogicIFEval into ten fine-grained difficulty intervals as follows:
>
> Interval 1: [12.00, 30.50]
>   Interval 2: [30.50, 37.00]
>   Interval 3: [37.00, 42.91]
>   Interval 4: [42.91, 49.00]
>   Interval 5: [49.00, 54.62]
>   Interval 6: [54.62, 62.25]
>   Interval 7: [62.25, 70.50]
>   Interval 8: [70.50, 83.00]
>   Interval 9: [83.00, 99.00]
>   Interval 10: [99.00, 233.50]
>
> And here are the performance of the representative models across the difficulty intervals:
> | Model | Interval 1 | Interval 2 | Interval 3 | Interval 4 | Interval 5 | Interval 6 | Interval 7 | Interval 8 | Interval 9 | Interval 10 | Overall |
> | :--- | :---: | :---: | :---: | :---: | :---: | :---: | :---: | :---: | :---: | :---: | :---: |
> | Claude Sonnet 4.0 | 0.857 | 0.884 | 0.744 | 0.762 | 0.860 | 0.738 | 0.643 | 0.605 | 0.512 | 0.372 | 0.697 |
> | Claude Sonnet 4.0 (No Thinking) | 0.833 | 0.698 | 0.605 | 0.381 | 0.488 | 0.381 | 0.333 | 0.279 | 0.233 | 0.163 | 0.439 |
> | DeepSeek R1 Distill Llama 70B | 0.690 | 0.535 | 0.535 | 0.286 | 0.395 | 0.286 | 0.190 | 0.209 | 0.140 | 0.093 | 0.336 |
> | GPT-4.1 | 0.810 | 0.791 | 0.605 | 0.524 | 0.674 | 0.595 | 0.500 | 0.395 | 0.465 | 0.302 | 0.566 |
> | GPT-5 | 0.976 | 0.907 | 0.884 | 0.881 | 0.953 | 0.833 | 0.833 | 0.721 | 0.767 | 0.744 | 0.850 |
> | Gemini 2.5 Flash | 0.833 | 0.791 | 0.605 | 0.643 | 0.581 | 0.571 | 0.333 | 0.372 | 0.326 | 0.209 | 0.526 |
> | Google Gemma 3 27B | 0.333 | 0.116 | 0.070 | 0.048 | 0.023 | 0.000 | 0.000 | 0.070 | 0.000 | 0.000 | 0.066 |
> | Qwen3 32B | 0.333 | 0.279 | 0.163 | 0.095 | 0.000 | 0.024 | 0.024 | 0.070 | 0.000 | 0.047 | 0.103 |
>
> We can clearly observe from the table that model performance decreases as the logical complexity of the instruction increases.
>
> The next table reports the total number of failure cases of the eight models evaluated in Figure 3 across error types (we add one more Arithmetic Errors) and difficulty levels:
> | Interval | Arithmetic Errors | Control Flow Misexecution | Instruction Misinterpretation | Misordered Execution | Missing Logic Elements | State Tracking Errors | Total |
> |:---------|:---:|:---:|:---:|:---:|:---:|:---:|:---:|
> | Interval 1 | 27 | 121 | 38 | 32 | 12 | 22 | 252 |
> | Interval 2 | 4 | 158 | 44 | 20 | 17 | 50 | 293 |
> | Interval 3 | 13 | 233 | 72 | 45 | 42 | 131 | 536 |
> | Interval 4 | 14 | 294 | 51 | 26 | 51 | 146 | 582 |
> | Interval 5 | 14 | 208 | 46 | 24 | 44 | 153 | 489 |
> | Interval 6 | 29 | 248 | 60 | 41 | 64 | 157 | 599 |
> | Interval 7 | 11 | 312 | 119 | 35 | 84 | 153 | 714 |
> | Interval 8 | 20 | 278 | 70 | 28 | 148 | 231 | 775 |
> | Interval 9 | 13 | 317 | 86 | 17 | 130 | 254 | 817 |
> | Interval 10 | 9 | 346 | 109 | 22 | 263 | 255 | 1004 |
>
> We observe that as instruction complexity increases:
>
> - The proportion of Control Flow Misexecution, Instruction Misinterpretation, Missing Logic Elements, and State Tracking Errors rises sharply. This suggests that current LLMs struggle primarily with understanding and executing complex logic, offering insight into where future modeling improvements should focus.
> - In contrast, Arithmetic Errors and Misordered Execution remain consistently low across all complexity levels. This indicates that models generally follow the instructed execution order correctly and Arithmetic errors do not interfere with evaluating instruction-following under complex logic in our benchmark.
>
> Hope these findings could address your concern.

---

> ### Author Response · Authors · 2025-11-15
> **Response to Weakness 2: Why We Prohibit LLMs Using Tools**
>
> Thanks for raising this. Please refer to [Common Response 1: Why We Prohibit LLMs Using Tools](https://openreview.net/forum?id=MzHBcYAats&noteId=5AgahLzkc2). We hope it could address your concern.

---

> ### Author Response · Authors · 2025-11-15
> **Response to Question 1: Further Explanation of Table 1 and Figure 3**
>
> Thanks for the question. The numbers in Table 1 are calculated at the **question level**:
> - **Output**: A question is counted if the model produces the correct outputs for all associated test cases.
> - **State**: A question is counted if the model produces the correct state trackers for all associated test cases.
> - **Both**: A question is counted if the model produces both correct outputs and state trackers for all associated test cases.
> In contrast, Figure 3 analyzes the failure modes and is computed at the **test-case level**, because different test cases under the same question may fail for different reasons. For example, the 28.1% state tracking for GPT-5 means that among all failed test cases of GPT-5, 28.1% are caused by state-tracking errors.
>
> We will make this distinction between question-level metrics (Table 1) and test-case-level failure distribution (Figure 3) clearer in the next revision.

---

> ### Author Response · Authors · 2025-11-15
> **Response to Question 2: Clairification of the "Average" and "Overall" of Table 1**
>
> Thanks for the question. The “Average” refers to **micro-averaging**. It is computed by summing the number of correctly solved questions across the three difficulty levels and dividing by the total number of questions (426). This metric reflects a model’s overall performance on the entire benchmark, which is why we use the term “Overall” in the caption. We will make it more clear in the next version.

---

> ### Author Response · Authors · 2025-11-15
> **Response to Question 3**
>
> Please see the [Response to Question 1](https://openreview.net/forum?id=MzHBcYAats&noteId=QiwesK2w2T). We will add these explanation to the next version.

---

> ### Author Response · Authors · 2025-11-21
>
> Dear Reviewer MrYZ,
>
> We have posted our responses to your questions and concerns. When you have time, please take a look. We hope our clarifications address your points. If you have any further questions or would like additional discussion, feel free to let us know.
>
> Best regards

---

> > ### Comment · Reviewer_MrYZ · 2025-11-24
> >
> > Thank you for the clarifications and further evaluations!
> >
> > Regarding W1, these new results are very relevant to your paper, but you need more than just observations and descriptions. If an explanation is not possible, then provide an interpretation in relation to a hypothesis. As I mentioned, please revise the scientific storyline of your paper to focus on dependencies and independence in instruction-following failure modes and logical complexity.
> >
> > Regarding W2, your response about tool use only partially addresses my concerns. Your paper still lacks a discussion of the theories and hypotheses concerning the capabilities of LLM behavior in instruction-following tasks.
> >
> > Please provide an updated manuscript that incorporates your highlighted changes (with respect to each review).

---

> ### Author Response · Authors · 2025-11-25
>
> Thanks again for the quick response. We'd like to further clarify the following points:
>
> - For Weakness 1, please see lines 426-435 and the Figure 4 in the **updated manuscript (in blue text)** for the **clear demonstration** of the relationship between the logical complexity and the failure modes.
>
> - For Weakness 2, our task is **indeed** evaluating the instruction following ability of LLMs for the following reasons:
>     1. The instructions we generate **explicitly specify the action required at each step**, removing the need for the model to infer what to do next. The model’s task is purely to follow and execute the given steps and adhere to the underlying logic.
>     2. Second, we perform rigorous data filtering to ensure that **each step of the instruction is simple enough for the model to execute directly**, without requiring additional external reasoning effort (see Section 3). As we shown in [our previous response to W1](https://openreview.net/forum?id=MzHBcYAats&noteId=RpoxC695Dp), the number of Arithmetic Errors is very low across the models, which shows that the each step is solvable and what challenges the model are the complex logic in the instructions.
>     3. Our work **is inspired by a famous instructon following work called [ComplexBench](https://arxiv.org/pdf/2407.03978)** (see Section Related Work), which investigates compositional constraint following, introducing logic structures such as sequential logic (the output must perform multiple tasks in sequence), and branching logic (the output must select different branches based on certain conditions). In contrast, the logical structures in our benchmark are significantly **more complex and diverse**. Moreover, the instructions generated by our proposed framework LogicIFGen possess three key advantages:
>         - **Verifiable** — each instruction is paired with executable gold labels;
>         - **Scalable** — instructions can be generated automatically from any code function;
>         - **Complexity-qualifiable** — instruction difficulty can be quantified based on the complexity of the underlying function.
>
> Thanks for your time to review our response and feel free to raise any quesitons. We would be more than happy to dive into a more in-depth discussion.

---

### Official Review · Reviewer_fakE · 2025-11-01

**Soundness:** 3
**Presentation:** 3
**Contribution:** 3
**Rating:** 6
**Confidence:** 4

**Summary:**

The paper proposes LogicIFGen and LogicIFEval. LogicIFGen is a method framework of generating natural language instructions based on code, and LogicIFEval is a benchmark derived by using the framework on hard coding problems. Then the authors tested the current LLMs on the benchmarks and shows it is very challenging for the current models.

**Strengths:**

- The paper studies an interesting problem of building complex natural language instructions from code and test the model's instruction following ability by using these code generated instructions.
- The paper's main pipeline of building these instructions is interesting and solid. The paper also did a decent job in collecting the coding problems which could be a contribution to the community.

**Weaknesses:**

- The naming is very confusing. Fundamentally LogicIFGen is the framework and LogicIFEval is derived from using this. But the naming made this very misleading.
- I think the analysis part is not solid enough. For example, some simple reasoning baselines such as Program-of-Thought should be tested and analyzed. Since the instruction is derived from code, I think in general the paper should consider the aspect of reasoning with code.

**Questions:**

- I would like to see how the current models w/ strong reasoning methods perform on this task.
- I'm curious how do the author view the difference between using natural language instructions vs directly using code as instructions and asking the model to follow the logic.

---

> ### Author Response · Authors · 2025-11-15
> **Response to the Benchmark Name**
>
> We follow the naming of IFEval (https://arxiv.org/abs/2311.07911). Maybe we need to consider changing it to “LogicIFBench”.

---

> ### Author Response · Authors · 2025-11-15
> **Response to Reasoning with Reasoning with Code.**
>
> Thanks for raising this. Please refer to [Common Response 1: Why We Prohibit LLMs Using Tools](https://openreview.net/forum?id=MzHBcYAats&noteId=5AgahLzkc2). We hope it could address your concern.

---

> ### Author Response · Authors · 2025-11-15
> **Response to Additional Results of Models w/ Strong Reasoning Methods**
>
> Thanks for the question. We may include additional evaluations of models with stronger reasoning techniques in the next version. At the current stage, however, our goal is to benchmark the **default** behavior of current frontier open-source and closed-source models. We believe this is the most general and informative setting to reflect their real-world ability without auxiliary enhancements. Exploring the effects of more advanced techniques is slightly beyond the scope of this work, and we expect LogicIFEval to serve as a strong testbed for such future investigations.

---

> ### Author Response · Authors · 2025-11-15
> **Response to the Difference Between Using Natural Language Instructions and Directly Using Code as Input**
>
> **Understanding the logical structures embedded in natural language is a more general and practically important capability**, as today’s agentic workflows and operational constraints are predominantly described using **natural language**. Like the Claude system prompts (https://www.dbreunig.com/2025/05/07/claude-s-system-prompt-chatbots-are-more-than-just-models.html), instructions today could contain description of complicated agentic model behaviour evolving condition, error handling, or function calling. Evaluating how well LLMs can follow such instructions is precisely the problem our work aims to address.

---

> ### Author Response · Authors · 2025-11-21
>
> Dear Reviewer fakE,
>
> We have posted our responses to your questions and concerns. When you have time, please take a look. We hope our clarifications address your points. If you have any further questions or would like additional discussion, feel free to let us know.
>
> Best regards

---

### Author Response · Authors · 2025-11-15
**Common Response 1: Why We Prohibit LLMs Using Tools & Motivation**

The reason we prohibit the model from using programming or external tools is that our goal is to evaluate the **instruction-following ability of LLMs**, rather than their **coding ability**. Instruction following is a more general and fundamental capability because, in most real-world scenarios, LLMs are expected to follow the logic embedded in natural language through text generation.

For example, a simple instruction such as “When the user asks something harmful to society, refuse and redirect the conversation to a safer topic” requires the model to follow the logic purely via natural-language behavior. More complex cases can be found in the Claude system instructions (https://www.dbreunig.com/2025/05/07/claude-s-system-prompt-chatbots-are-more-than-just-models.html), which describe sophisticated agentic behaviors involving conditions, error handling, and reflection. Importantly, the execution of these behaviors is still realized primarily through text generation.

We acknowledge that real-world instructions may involve guidance on when and how to use external tools, and that LLMs are often expected to call tools in such scenarios. However, **allowing tool use during evaluation would make the evaluation both complicated and unfair**: there exists a vast number of possible tools, and different models have unequal tool-calling capabilities. Our objective is to **isolate and evaluate the instruction-following ability of LLMs**, without introducing potential confounding factors. To achieve this, we perform rigorous data filtering to guarantee that every substep of the instruction in our benchmark is simple enough for the model to complete internally, without the help of external tools (see Section 3).

---

### Author Response · Authors · 2025-12-02

Dear Area Chairs,

Below is a brief summary of the main concerns raised by the reviewers, along with pointers to our corresponding responses and revisions to assist your review:

---

### Reviewer fakE
**Main Concern**: The authors should consider reasoning with code

**Our Response**: Please see our response on why we prohibit LLMs using tools ([link](https://openreview.net/forum?id=MzHBcYAats&noteId=5AgahLzkc2)).

---

### Reviewer MrYZ
**Main Concern 1**: Missing analysis connecting LLM behavior and logical complexity of instructions

**Our Response**: We have added a detailed analysis of failure modes as instruction complexity increases. This includes a breakdown of how the logical complexity affects model errors in Response to Weakness 1 ([link](https://openreview.net/forum?id=MzHBcYAats&noteId=RpoxC695Dp)) and in the updated manuscript (lines 426–435, Figure 4; newly added text in blue). These additions clearly demonstrate the relationship between logical complexity and LLM failure patterns.


**Main Concern 2**: Lack of discussion on the motivation behind the introduced dataset

**Our Response**: Please see the detailed explanation of the motivation ([link](https://openreview.net/forum?id=MzHBcYAats&noteId=5AgahLzkc2)) and why our benchmark is targeted to evaluate the LLMs' instruction following ability to complex logic ([link](https://openreview.net/forum?id=MzHBcYAats&noteId=cLaiS6sj8L)).

---
### Reviewer xW3R
**Main Concern**: Relationship between our task and code execution is unclear

**Our Response**: We have provided a detailed explanation of how our task (logical instruction following) differs from code execution and code simulation benchmarks ([link](https://openreview.net/forum?id=MzHBcYAats&noteId=VHskH4GJJs)). These discussions have been integrated into the Related Work section of the updated manuscript (text added in blue).

---

We appreciate the reviewers’ valuable concerns and have carefully addressed them in our responses and revisions. We are also sorry for the additional time and efforts needed to review our rebuttal due to the system reset.

Thank you,

Authors

---

### Meta-Review · Area_Chair_Ks51 · 2026-01-01

**Summary:**

This paper proposes the LogicIFGen framework and LogicIFEval benchmark to evaluate LLMs on complex logical instructions derived from code, but the decision is weighed down by reviewer concerns regarding the depth of the failure mode analysis and the theoretical justification for isolating logical instruction following from tool use.

**Reviewer Concerns:**

The rebuttal successfully clarified the distinction between this work and existing code execution benchmarks (addressing Reviewer xW3R's novelty concern), but Reviewer MrYZ's request for a theoretical framework explaining why specific failure modes correlate with complexity remains outstanding.

**Reviewer Scores:**

Reviewer xW3R may raise their score upon recognizing the clear distinction from code simulation tasks, whereas Reviewer MrYZ may likely maintain their lower score due to the persistent lack of a scientific hypothesis grounding the empirical observations.

---

### Decision · Program_Chairs · 2026-01-26

Reject